# Fragmentation and aggregation of cyanobacterial colonies

**Yuri Z Sinzato[1]\*, Robert Uittenbogaard[2], Petra M Visser[3], Jef Huisman[3], Maziyar Jalaal[1,4]\***

[1]Van der Waals-Zeeman Institute, Institute of Physics, University of Amsterdam, Amsterdam, Netherlands; [2]Hydro-Key Ltd, Haelen, Netherlands; [3]Department of Freshwater and Marine Ecology, Institute for Biodiversity and Ecosystem Dynamics, University of Amsterdam, Amsterdam, Netherlands; [4]Department of Applied Mathematics and Theoretical Physics, University of Cambridge, Cambridge, United Kingdom

## eLife Assessment

With the goal of investigating the assembly and fragmentation of cellular aggregates, this manuscript examines cyanobacterial aggregates in a laboratory setting. This quantitative investigation of the conditions and mechanisms behind aggregation is an **important** contribution as it yields a basic understanding of natural processes and offers potential strategies for control. The combination of computational and experimental investigations in this manuscript provides **convincing** support for the role of shear on aggregation and fragmentation.

**\*For correspondence:**
y.z.sinzato@uva.nl (YZS);
m.jalaal@uva.nl (MJ)

**Abstract** Fluid flow has a major effect on the aggregation and fragmentation of bacterial colonies. Yet, a generic framework to understand and predict how hydrodynamics affects colony size remains elusive. This study investigates how fluid flow affects the formation and maintenance of large colonial structures in cyanobacteria, using an experimental technique that precisely controls hydrodynamic conditions. We performed experiments on laboratory cultures and lake samples of the cyanobacterium *Microcystis*, while their colony size distribution was measured simultaneously by direct microscopic imaging. We demonstrate that extracellular polymeric substances (EPS)-embedded cells formed by cell division exhibit significant mechanical resistance to shear forces. However, at elevated hydrodynamic stress levels (exceeding those typically generated by surface wind mixing), these colonies experience fragmentation through an erosion process. We also show that single cells can aggregate into small colonies due to fluid flow. However, the structural integrity of these flow-induced colonies is weaker than that of colonies formed by cell division. We provide a mathematical analysis to support the experiments and demonstrate that a population model with two categories of colonies describes the measured size distributions. Our results shed light on the specific conditions wherein flow-induced fragmentation and aggregation of cyanobacteria are decisive and indicate that colony formation under natural conditions is mainly driven by cell division, although flow-induced aggregation could play a role in dense bloom events. These findings can be used to improve prediction models and mitigation strategies for toxic cyanobacterial blooms and also offer potential applications in other areas, such as algal biotechnology or medical settings where the dynamics of biological aggregates play a significant role.

**eLife digest** Anyone who has seen a lake turn green in summer has witnessed the visible sign of a cyanobacterial bloom. Cyanobacteria are photosynthetic microorganisms, and some species are toxic and cause major environmental problems. A common nuisance species is *Microcystis*, which lives in colonies – clusters of cells held together by a slimy material called extracellular polymeric substance, which acts like a biological glue.

These colonies can reach large dimensions and thus float faster than single cells and receive more light, making blooms harder to control. They can grow through cell division, in which newly formed cells remain attached, or through aggregation, in which separate cells or colonies collide and stick together. Water movement also plays a crucial role. Fluid flow can promote aggregation, but it can also generate shear forces that break colonies apart. So far, it was unclear which of these competing processes dominates under realistic lake conditions or during mitigation efforts such as artificial mixing.

To find out how fluid flow influences the size of *Microcystis* colonies, Sinzato et al. studied colonies from both laboratory cultures and field samples under controlled shear flow. They quantified changes in colony size and compared the results with a mathematical model that incorporates both fragmentation (breaking apart) and aggregation (sticking together) processes.

The results showed that *Microcystis* colonies are surprisingly mechanically robust and are not readily fragmented by wind-driven or artificial mixing. Moreover, colonies formed through cell division were significantly stronger than those formed through aggregation, most likely because aggregated cells do not have enough sticky material to form a strong bond.

Colony size of cyanobacteria affects buoyancy, light exposure and bloom development, which, in turn, influences the effectiveness of bloom forecasting and control strategies. The findings of Sinzato et al. shed light on the mechanisms underlying colony formation and could help improve predictive models and mitigation strategies for harmful cyanobacterial blooms. They may also assist engineers in understanding how dense cell aggregates interact with fluid flow in algal bioreactors.

More broadly, the combined approach of flow studies, imaging, and modeling used in this study could be applied to other biological aggregates exposed to flow, such as marine snow. However, further work is needed to test this framework across a wider range of species and environmental conditions.

## Introduction

Many environmental and pathogenic bacteria exhibit colonial forms of life, either as surface biofilms (*Stoodley et al., 2002*; *Flemming and Wingender, 2010*) or non-attached aggregates (*Katharios-Lanwermeyer et al., 2014*; *Trunk et al., 2018*; *Martínez-Calvo et al., 2022*). In most situations, these colonies are subjected to hydrodynamic stresses, which may lead to flow-induced fragmentation and aggregation (*Burd and Jackson, 2009*). These physical processes, combined with biological growth, determine the fate of aggregates and the microbial lifestyle. Nonetheless, despite their abundance and fundamental importance, a generic framework to study the effects of fluid flow on bacterial colonies, as well as their fragmentation and aggregation across scales, is still in its infancy (*Drescher et al., 2013*; *Rusconi et al., 2014b*; *Rusconi et al., 2014a*; *Atis et al., 2019*). Here, we focus on colonies of bloom-forming cyanobacteria and ask the question: how do flow-induced aggregation and fragmentation regulate dynamic changes in colony size?

Toxic cyanobacterial blooms are a major nuisance for freshwater ecosystems worldwide (*Huisman et al., 2018*). These blooms often lead to increased turbidity, cause harm to other aquatic organisms, and deteriorate water quality. Recent evidence indicates that the severity and frequency of cyanobacterial blooms are increasing due to eutrophication and climate change (*Paerl and Huisman, 2009*; *Visser et al., 2016b*; *O'Neil et al., 2012*; *Ho et al., 2019*). Consequently, water management authorities show growing interest in techniques to prevent, monitor, and control these toxic blooms (*Ibelings et al., 2016*). A particular effort is being taken to combat blooms of *Microcystis*, a non-motile colony-forming freshwater cyanobacterial genus that dominates many aquatic ecosystems worldwide (*Xiao et al., 2018*; *Harke et al., 2016*). Fluid dynamics is at the heart of *Microcystis* bloom formation, but is also used in some of the methods to control them. A large colony size and positive buoyancy

give *Microcystis* a higher average light dose in comparison to non-buoyant algae (*Visser et al., 1997*; *Huisman et al., 2004*). Control methods such as deep artificial mixing have been successfully applied in several lakes to disrupt this competitive advantage by increasing the turbulence level (*Visser et al., 2016a*). Yet, the mechanisms of *Microcystis* colony formation, and, in particular, the impact of fluid flow on the formation of colonies remain poorly understood.

*Microcystis* cells are held together by a layer of extracellular polymeric substances (EPS), facilitating the formation of large colonies of up to a millimeter in diameter with diverse morphological arrangements (*Le et al., 2022*; *Liu et al., 2018*; *Via-Ordorika et al., 2004*). Two main mechanisms for colony formation have been proposed (*Xiao et al., 2018*): (i) cell division, i.e., clonal expansion of cells which remain attached by their secreted EPS layer after division and (ii) aggregation, i.e., initially independent cells/colonies collide and adhere to each other. Sequencing of *Microcystis* colonies has revealed a much higher genetic diversity between independent colonies than within colonies, thus supporting the hypothesis that colony formation under natural conditions is primarily driven by cell division (*Pérez-Carrascal et al., 2021*). In contrast, cell aggregation (sometimes also called cell adhesion) can promote a rapid increase in colony size beyond the limit set by division rates and may explain sudden rises in colony size in late bloom periods (*Xiao et al., 2017*; *Zhu et al., 2015*; *Li et al., 2013*). Aggregation rates depend on the stickiness of the colonies, which in turn is controlled by the EPS composition, pH, and ionic composition of water (*Leussen, 1988*; *Kiørboe and Hansen, 1993*; *Verspagen et al., 2006*). In particular, divalent cations such as $Ca^{2+}$ can bridge negatively charged functional groups in EPS and therefore increase stickiness (*Xu et al., 2016a*; *Wang et al., 2011b*; *Sato et al., 2017*). It has been shown that high levels of $Ca^{2+}$ enhance cell aggregation in *Microcystis* cultures (*Chen and Lürling, 2020*). Moreover, cell aggregation can provide a fast defense against grazing (*Yang et al., 2006*). Fluid flow plays an important role in cell aggregation by regulating the collision frequency between cells or colonies (*Burd and Jackson, 2009*). In addition, fluid flow also influences colony formation through fragmentation, potentially serving as a mechanism for colony reproduction (*Day et al., 2022*) to overcome the growth limitations of large colonies (*Feng et al., 2020*). Beyond modulating colony size through aggregation and fragmentation, fluid flow has also been proposed as a driving factor in changes in colony morphology and growth rate (*Li et al., 2018*; *Wilkinson et al., 2016*; *Xiao et al., 2016*).

Flow-induced alterations in colony size and morphology impact the vertical migration and grazing resistance of colonies, thereby influencing the success of the *Microcystis* population. This bears significance for forecasting models which have often assumed a fixed colony size (*Aparicio Medrano et al., 2013*; *Rowe et al., 2016*; *Ranjbar et al., 2021*). Information on colony size is also important for bloom control via artificial mixing systems, which rely on estimates of colony flotation velocity, determined by

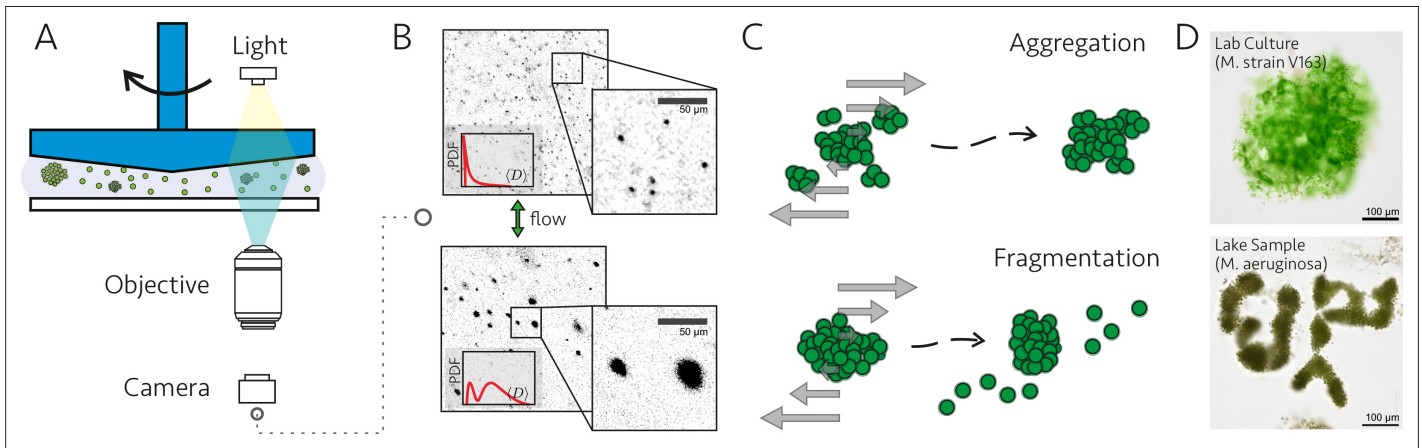

**Figure 1.** Methodology used to observe effects of fluid flow on *Microcystis* colonies. (**A**) Experimental setup consisting of a (cone-and-plate) controlled flow setup combined with inverted microscopy. The conical upper surface was rotated by a rheometer head, while the stationary glass slide below the sample allowed optical access for the microscope. (**B**) Examples of microscopy images. Colony size distributions were calculated after image processing of the captured frames. (**C**) Changes in size distributions (and other complementary measurements) over time were used to identify aggregation and fragmentation of cyanobacterial colonies. (**D**) The majority of the measurements were conducted using a laboratory culture of *Microcystis* strain V163. Colonies collected from Lake Gaasperplas (Netherlands), dominated by the morphospecies *Microcystis aeruginosa*, were also used.

both colony shape and size (*Visser et al., 2016a*; *Nakamura et al., 1993*). Moreover, it remains uncertain whether mixing systems, such as bubble plumes, can enhance their efficiency by fragmenting colonies or whether artificial mixing may inadvertently promote aggregation. Colony fragmentation has been observed in grid-stirred (*O'Brien et al., 2004*) and propeller-mixed tanks (*Li et al., 2018*) at energy dissipation rates comparable to those in bubble plumes (*Lai and Socolofsky, 2019*). However, these studies reported averaged dissipation rates, while local values near the fast-moving surfaces can be orders of magnitude higher (*Peter et al., 2006*). Furthermore, the size distribution of the colony fragments has not been characterized.

In the present study, we investigated to what extent *Microcystis* colony fragmentation and/or aggregation occurs at levels of turbulence observed for naturally or artificially mixed lakes. To this end, colonies of a laboratory culture of *Microcystis* were subjected to various intensities of shear flow (*Figure 1*). We identified the mechanism of colony fragmentation using direct measurements of dynamic changes in colony size distribution under controlled laboratory conditions and compared the characteristics of colonies formed by aggregation and cell division. Furthermore, the fragmentation behavior of the laboratory culture was compared with field samples of *Microcystis* spp. We also developed a numerical model based on a two-category discrete population to simulate changes in the size of *Microcystis* colonies via aggregation and fragmentation. Finally, we provide a phase map based on flow characteristics and cyanobacterial abundance to highlight different regimes in which fluid flow plays a significant role in cyanobacterial colony formation.

## Results

### Fragmentation of large *Microcystis* colonies occurs through erosion

A culture of *Microcystis* strain V163 was filtered to collect large colonies, and colonies on the filter were subsequently re-dispersed in fresh BG-11 medium up to the desired total biovolume fraction $\phi$ (Materials and methods). The flow was generated by a cone-and-plate shear operating in an inertial regime, and it was characterized by the average energy dissipation rate $\dot{\varepsilon}$, hereafter called dissipation rate (Materials and methods). The size distribution of the colonies was measured over time by microscopic imaging of the suspension through the transparent lower surface (*Figure 1*). The colony size is described here by its relative diameter $l = d/d_1$, where $d$ is the Feret's diameter (maximum distance between two boundary points of the colony) and $d_1$ is the single-cell diameter. The number of cells in a colony, $N_c$, was calculated from its relative diameter via the relation $N_c = l^f$, where $f = 2.09$ is the average fractal dimension measured for colonies of *Microcystis* strain V163 (Materials and methods). The size distribution was computed as the normalized biovolume distribution $n(l)$, i.e., the number of cells in colonies in size bin $\{l; l + \Delta l\}$, divided by the total number of cells in the suspension and the bin width.

The results show that the initial size distribution had a bimodal shape, with a subpopulation of small colonies peaking around $l \sim 1 - 2$ (single cells, dimers, and trimers) and a subpopulation of large colonies peaking around $l \sim 20 - 40$ (*Figure 2A*). The cutoff size between the two subpopulations is defined here as the minimum of the bimodal distribution. Some small colonies ($l \sim 1 - 2$), up to 20% of the total biovolume, were always present in the initial distribution even after filtration, due to pre-stirring required to keep the suspension homogeneous. However, medium-sized colonies ($l = 4 - 8$) accounted for a negligible fraction of the distribution.

When subjected to an intense dissipation rate ($\dot{\varepsilon} = 5.8 \, \mathrm{m^2/s^3}$), the large colonies of *Microcystis* strain V163 displayed considerable fragmentation, with their median diameter (50th percentile of the biovolume distribution of the large colonies) decaying to about half of the initial value after a few hours (*Figure 2B*). In contrast, the median diameter of small colonies showed little change in size and remained below $l \leq 2$ during the entire experiment. The size distribution after 1 hr indicated that most colonies larger than $l \gtrsim 30$ were fragmented, leading to a rise in the subpopulation of small colonies (*Figure 2C*). Notably, the bimodal shape of the size distribution was preserved, with very few medium-sized colonies. This suggests that aggregation of single cells into larger colonies is negligible under this intense shear flow condition, and the main effect of the shear flow on the size distribution is the decrease in size of the large colonies concomitant with an increase in the fraction of small colonies. After a few hours of shear, about 80% of the total biovolume was composed of small colonies (*Figure 2D*). This time scale was much shorter than the cell division time, such that changes

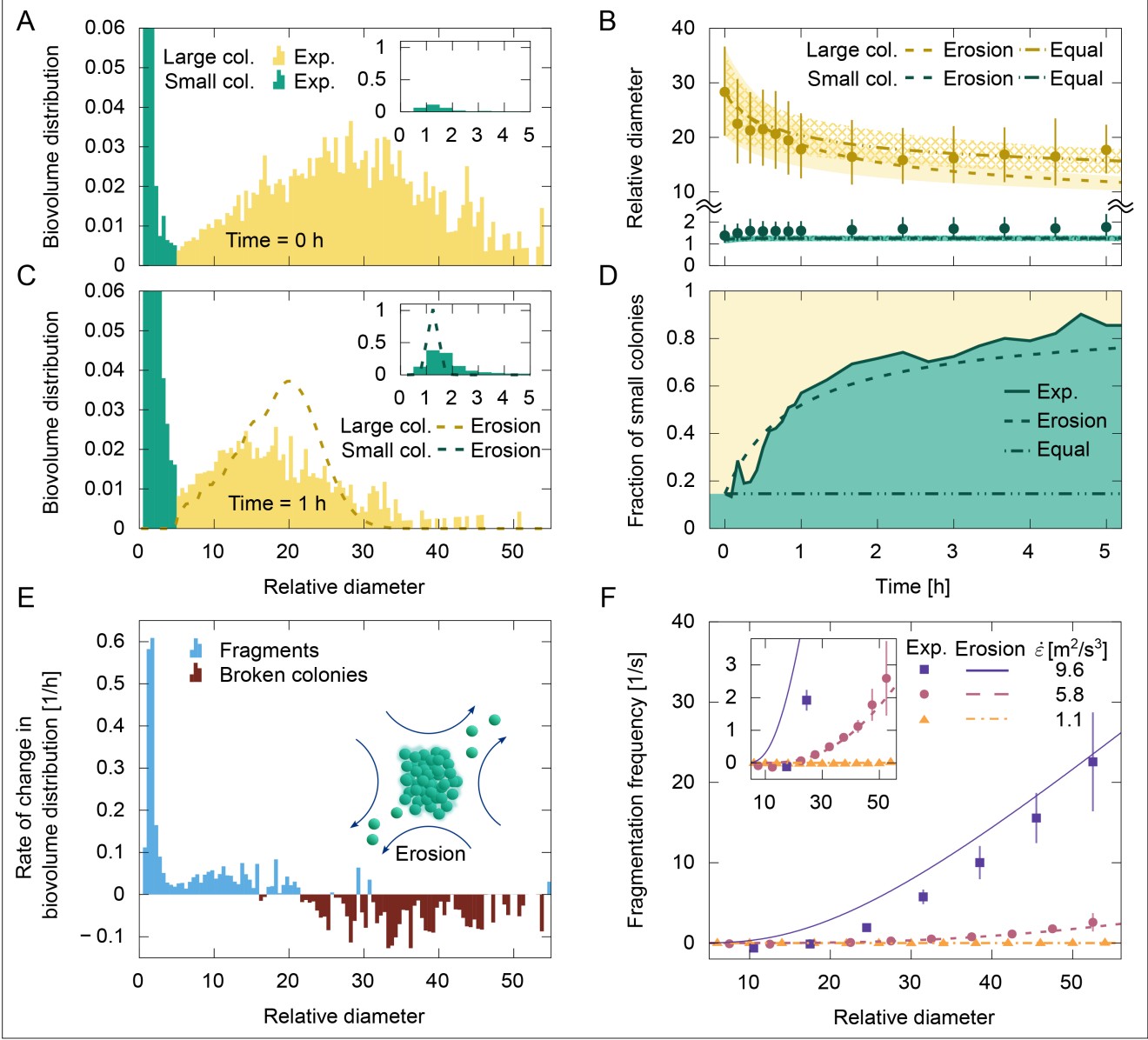

**Figure 2.** Kinetics of fragmentation of *Microcystis* strain V163 colonies under cone-and-plate shear flow. The laboratory culture was filtered to select mainly large colonies, and the total biovolume fraction was adjusted to $\phi = 10^{-4}$. Suspensions were subjected to an intense dissipation rate ($\dot{\varepsilon} = 5.8\,\mathrm{m^2/s^3}$) in panels **A–E**. (**A**) The initial size distribution of colonies, expressed as biovolume fraction of the relative colony diameter (normalized by single-cell diameter). The size distribution had a bimodal shape, with large colonies ($l > 5$, in yellow) and small colonies ($l \leq 5$, in green, composed mostly of single cells, dimers, and some trimers, as depicted by the inset). (**B**) Median diameter of small and large colonies as a function of time. Symbols indicate the experimental data, while the lines indicate the predictions from the population model given by *Equation 1* with an erosion (dashed) or equal-fragment (dot-dash) hypothesis for large division-formed colonies. Small aggregated colonies follow the equal-fragment hypothesis. Bars and shaded region indicate limits of 25th and 75th percentiles. (**C**) Most large colonies have been fragmented after 1 hr of shear flow, but the bimodal shape of the colony size distribution remained. (**D**) Biovolume fraction of small colonies (i.e. biovolume of small colonies over the total biovolume) as a function of time. A shift is observed from large to small colonies, captured well by the erosion model, but not by the equal-fragment model. (**E**) The rate of change in biovolume distribution at *t*=0 hr. Negative values indicate loss of colonies by fragmentation, while positive values indicate newly created fragments. The distribution suggests an erosion mechanism, as depicted by the cartoon inside the plot. (**F**) Fragmentation frequency as a function of the relative diameter of colonies for three values of dissipation rate. The inset shows details for moderate dissipation rates. Lines indicate the predictions by *Equation 5*. Bars indicate the uncertainty in the fragmentation frequency propagated from the uncertainty of the concentration distribution (Appendix 1). Best fit parameters: $\alpha_1 = 0.023$, $S_1 = 0.034$, $q_1 = 4.5$; erosion: $S_2 = 31$, $q_2 = 4.1$; equal fragments: $S_2 = 33$, $q_2 = 6.3$.

in the colony size distribution by division could be neglected within the experimental time (*Huisman et al., 2004*).

Direct measurement of the fragment distribution function $g(l, l')$ (i.e. the frequency of fragments with size $l$ arising from the fragmentation of a colony of size $l'$) is an experimental challenge (*Flesch et al., 1999*; *Vankova et al., 2007*; *Håkansson, 2020*). Individual fragmentation events could not be visualized as the residence time of colonies inside the field of view was too short. Instead, a suitable model can be identified from the rate of change in the biovolume distribution (*Figure 2E*). A negative rate was observed for large colonies in the range $l \gtrsim 25$, indicating a loss in that size range by fragmentation. In contrast, the newly created fragments appeared as positive peaks, with a distinct peak close to the single-cell size. This result is compatible with an erosion mechanism, in which small fragments with one or a few cells are detached from the outer layer of the colony. For an ideal binary erosion, a fragmentation event of a colony with $i$ cells results in one cell being eroded from the surface of the colony with a complementary fragment of $i - 1$ cells remaining. Small deviations from an ideal binary erosion are observed in *Figure 2E*, with most eroded fragments lying in the range $1 \leq l \leq 2$. Nevertheless, ideal erosion provides a good approximation for the results. Accurate identification of complementary fragments from the rate of change in the biovolume distribution was more difficult, because the broad initial distribution of our suspension led to a broad distribution of complementary fragments. In this case, the rate of change in the biovolume distribution shifted from a net loss of eroded colonies at larger sizes ($l \gtrsim 25$) to a net gain of complementary fragments at intermediate sizes ($5 \lesssim l \lesssim 20$).

Assuming an ideal binary erosion mechanism, the fragmentation frequency $\kappa(l)$ was computed from the temporal dependence of the biovolume distribution (Materials and methods). Here, $\kappa(l)$ is defined as the average number of fragmentation events (loss of a single cell) that one colony of size $l$ suffers per unit of time. The fragmentation frequency of *Microcystis* strain V163 was calculated as a function

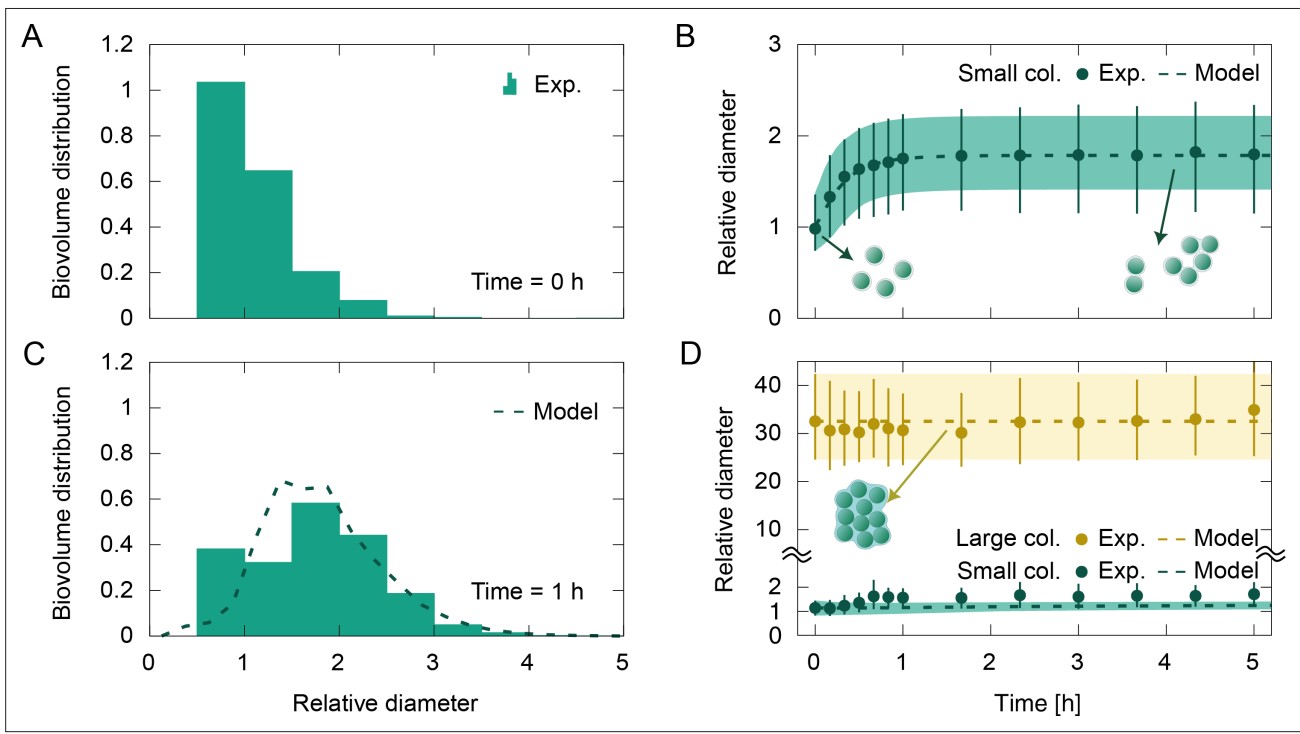

**Figure 3.** Kinetics of aggregation for a single-cell suspension of *Microcystis* strain V163 at a moderate dissipation rate of $\dot{\varepsilon} = 0.019 \, \mathrm{m^2/s^3}$ and a total biovolume fraction of $\phi = 10^{-4}$. (**A**) Initial size distribution of the suspension as a function of the relative diameter, composed mostly of single cells. (**B**) Median diameter of colonies formed by aggregation of single cells as a function of time. Symbols indicate the experimental data, while the lines indicate the predictions from the population model given by *Equation 1* with an equal-fragment hypothesis for small aggregate colonies. (**C**) After 1 hr of shear flow, the size distribution shifted toward slightly larger diameters. (**D**) Time behavior of a suspension of large division-formed colonies under the same moderate dissipation rate and total biovolume fraction. The bimodal size distribution is separated into large (yellow) and small (green) colonies. Dashed lines and shaded regions indicate predictions from the population model given by *Equation 1*. Bars and shaded region in panels B and D indicate limits of 25th and 75th percentiles. Best fit parameters: $\alpha_1 = 0.023$, $S_1 = 0.034$, $q_1 = 4.5$, $S_2 = 31$, $q_2 = 4.1$.

of colony size for three values of dissipation rate (*Figure 2F*). The results show that the fragmentation frequency increases with colony size. Furthermore, an increase in dissipation rate sharply intensifies the erosion of colonies, with a two-orders-of-magnitude increase in $\kappa(l)$ when $\dot{\varepsilon}$ varies from 1.1 to $9.6\,\mathrm{m^2/s^3}$. The intensified fragmentation frequency also reduces the time required for the biovolume distribution to reach a steady state (*Appendix 1—figure 1*). It is also noticeable that some of the large colonies (5–10% of the total biovolume) do not fragment to single cells but remain as large colonies with relative diameter of 10–20 even after 5 hr of very high dissipation rate ($\dot{\varepsilon} = 9.6\,\mathrm{m^2/s^3}$), indicating variability in the mechanical strength of colonies, with a subset presenting an exceptional resistance in comparison to the average behavior (*Appendix 1—figure 1E and F*).

## Aggregation-formed colonies are less resistant than division-formed colonies

In addition to inducing fragmentation, a turbulent flow may also increase the collision frequency of colonies. If the collisions are adhesive and the suspension is concentrated enough, then new colonies will be formed by aggregation. We subjected a single-cell suspension to the cone-and-plate shear flow to investigate whether aggregation can be a dominant colony-forming mechanism compared to cell division. Cultures of the *Microcystis* strain V163 were filtered to obtain a single-cell suspension, and the total biovolume fraction was adjusted with fresh BG-11 medium (Materials and methods). A typical initial biovolume fraction distribution is presented in *Figure 3A*. In addition to single cells, some small colonies up to $l = 3$ were present in the filtered suspension due to imperfect filtering and/or aggregation during the pre-stirring. The size distribution was unimodal (single peak, *Figure 3A*), in contrast with the bimodal distribution (double peak, *Figure 2A*) in the suspension of division-formed colonies. After about 1 hr at a moderate dissipation rate ($\dot{\varepsilon} = 0.019\,\mathrm{m^2/s^3}$) and a total biovolume fraction of $\phi = 10^{-4}$, the single-cell suspension showed aggregation of cells into small colonies, with a steady-state median diameter of $l = 1.8$ (*Figure 3B and C*). Note that cell division can be neglected as the measurement time was much shorter than the division time. An increase in the collision rates due to a denser single-cell suspension led to an increase in the median diameter, although the increase was minor. At a high total biovolume fraction ($\phi = 5 \cdot 10^{-4}$), a steady-state median diameter of $l=24$ was reached, with a negligible fraction of colonies above $l \gtrsim 5$ (*Appendix 1—figure 2C*). Therefore, the aggregation of single cells is likely limited by weak bonds between cells. At the steady-state regime of median diameter, the newly formed aggregates with a large diameter would be quickly fragmented back into smaller parts.

A suspension of division-formed colonies under the same moderate dissipation rate ($\dot{\varepsilon} = 0.019\,\mathrm{m^2/s^3}$) did not suffer noticeable fragmentation (*Figure 3D*). The culture was grown in a quiescent medium, without shaking, thus the colonies are assumed to have formed solely by cell division. The erosion of division-formed colonies was negligible, as the fraction of small colonies remained below 20% (*Appendix 1—figure 3B*). Furthermore, the nearly constant median diameter of the large colonies (observed also for other total biovolume fraction values, *Appendix 1—figure 3*) shows that the division-formed colonies did not aggregate at this moderate dissipation rate.

The results above indicate that bonds formed during flow-induced aggregation (whether between two single cells or two division-formed colonies) are weak and cannot withstand intense dissipation rates. In contrast, the bonds between cells within a single division-formed colony are significantly more resistant. This difference in mechanical strength of the cellular binding in aggregation-formed versus division-formed colonies can be attributed to the structure of the EPS layer that surrounds the colonies. Aggregated colonies would have intercellular gaps (smaller than the cell size) that are not filled with EPS. In contrast, cell division allows enough time for the gaps to be filled with secreted EPS, effectively increasing the bond strength between cells (*Figure 3B and D*). However, we could not confirm this hypothesis visually, as the EPS layer in *Microcystis* strain V163 is too thin (<1 μm) for a clear identification using optical microscopy.

## Dynamical changes in colony size modeled by a two-category distribution

Our experimental results for colonies of *Microcystis* strain V163 (*Figure 3*) suggest the presence of two categories of colonies: single cells and small aggregates (denoted as $C_1$) and large colonies formed by cell division ($C_2$). The total biovolume distribution $n(l)$ is given by the sum of both distributions

$n_1(l) + n_2(l)$, normalized such that $\int_0^\infty n(l)\,\mathrm{d}l = 1$. Within times shorter than the cell division time, the biovolume distribution of each colony category is described by a population model (*Leussen, 1988*):

$$\frac{\partial n_i}{\partial t}(l, t) = A_i(l, t) + F_i(l, t) + T_i(l, t) , \tag{1}$$

where $A_i(l, t)$ and $F_i(l, t)$ describe the aggregation and fragmentation rates as a function of colony size for each of the two categories ($i \in [1, 2]$), and $T_i(l, t)$ is a transfer rate of biovolume between the two categories. The turbulent shear-induced collisions between colonies shift the distribution toward larger aggregates, and the aggregation rate is given by:

$$A_i(l, t) = \frac{6\,\phi\,\alpha_i}{\pi} \left[ \frac{1}{2} \int_0^l \beta(l', l_o) \left(\frac{l}{l' l_o}\right)^{2f-1} n_i(l', t)\, n_i(l_o, t)\, \mathrm{d}l' \right.$$
$$\left. - n_i(l, t) \int_0^\infty \beta(l, l') \frac{1}{l'^f} n_i(l', t)\, \mathrm{d}l' \right] \tag{2}$$

where $l_o = (l^f - l'^f)^{1/f}$. The collision frequency is described by the kernel $\beta(l, l')$ and for a turbulent shear flow, we have $\beta(l, l') = 0.163\,(l + l')^3 \sqrt{\dot{\varepsilon}/\nu}$, where $\nu$ is the kinematic viscosity of the fluid (*Saffman and Turner, 1956*). The ratio of aggregative collisions over total collisions is given by the stickiness parameter $\alpha_i$. For the category $C_1$ (single cells and small aggregates), the stickiness is positive, $\alpha_1 > 0$. In contrast, we neglect the aggregation of $C_2$ colonies (collisions between $C_2$–$C_2$ and $C_1$–$C_2$), since the bonds formed by aggregation would be weaker and quickly rupture again, such that $\alpha_2 = 0$. This assumption is corroborated by the absence of aggregation between large colonies in our experiments (*Figure 3D*).

The colony formation will be limited by fragmentation if the colony exceeds a critical size $l_i^*$, given by:

$$l_i^* = \left(\frac{\dot{\varepsilon}}{S_i}\right)^{-q_i/f} , \tag{3}$$

where $S_i$ and $q_i$ are constitutive parameters that depend on the aggregate structure and bond strength, with a lower bound at $l_i^* \geq 1$ (*Bäbler et al., 2008*; *Zaccone et al., 2009*). We assume that aggregation-formed colonies have a small critical size, as the overlap between the EPS layer of each cell is narrow, leading to weak bonds. Meanwhile, division-formed colonies produce sufficient EPS to fill the space between cells, thus increasing the resistance to hydrodynamic stress. The fragmentation rate is given by a first-order kinetic equation (*Bäbler et al., 2008*):

$$F_i(l, t) = -\kappa_i(l)\, n_i(l, t) + \int_l^\infty g_i(l, l')\, \kappa_i(l')\, n_i(l', t)\, \mathrm{d}l' , \tag{4}$$

where $\kappa_i(l)$ is the fragmentation frequency, which is a function of the probability distribution function of the hydrodynamic stress. Assuming a Gaussian distribution, the fragmentation frequency is (*Bäbler et al., 2008*):

$$\kappa_i(l) = \sqrt{\frac{2\,\dot{\varepsilon}}{15\,\pi\,\nu}} \frac{\exp(-M_i^2)}{\mathrm{erf}(M_i)} , \tag{5}$$

where $M_i = \sqrt{15/2}\,(l/l_i^*)^{-f/2q_i}$. Due to the scarcity of experimental data in the literature, simplified models are usually assumed for the fragment distribution function $g_i(l, l')$, such as erosion, equal fragments, or a uniform distribution (*Vanni, 2000*). Assuming a binary fragmentation (i.e. two fragments per fragmentation event), the fragment size distribution can be written in the form $g_i(l, l') = \delta(l - l_a) + \delta(l - l_b)$, where $l_a$ and $l_b$ are the sizes of fragments. Based on the results from division-formed colonies under intense dissipation rate (*Figure 2E*), an ideal erosion mechanism is adopted for category $C_2$ colonies, such that $g_2(l, l') = \delta(l - 1) + \delta(l - \sqrt[f]{l'^f - 1})$. In contrast, the experimental results for the small colonies (*Figure 3C*) show a wider size distribution of eroded fragments which cannot be captured by an ideal erosion mechanism, so an equal-fragment mechanism is adopted for category $C_1$ colonies, i.e., $g_2(l, l') = 2\delta(l - l'/\sqrt[f]{2})$. The size distributions of $C_1$ and $C_2$ colonies are

coupled through a transfer rate of biovolume $T_i(l,t)$. We propose that the cells eroded from $C_2$ colonies are transferred to category $C_1$, such that:

$$T_1(l,t) = -T_2(l,t) = \delta(l-1) \int_l^\infty \kappa_2(l') \, n_2(l',t) \, \mathrm{d}l' \,, \qquad (6)$$

while the complementary fragments of $C_2$ colonies remain in that category.

We solved the balance **Equation 1** using a zero-order discrete population balance, or sectional method, in which the size distribution is described by uniformly sized bins where the size distribution is evaluated at a representative size centered on the bin (**Vanni, 2000**; **Kumar and Ramkrishna, 1996**). The simulations were initialized with the size distributions at $t = 0$ obtained in our experiments. The two pairs of parameters for critical size ($\{q_1, S_1\}$ and $\{q_2, S_2\}$) and the stickiness parameter ($\alpha_1$) were used to fit the model to the experimental data (Materials and methods).

For an initial suspension composed mainly of large colonies at an intense dissipation rate ($\dot{\varepsilon} = 5.8 \, \mathrm{m}^2/\mathrm{s}^3$), the model prediction of a decrease in the median size of $C_2$ colonies up to a steady-state diameter is in good agreement with the experimental findings (**Figure 2B**). Moreover, the erosion mechanism accurately predicts the biovolume transfer from large to small colonies (**Figure 2D**). In comparison, the hypothesis of equal-fragment formation for $C_2$ colonies predicts that all fragments remain within the large colony size category ($l > 5$) during the first 5 hr of experiments; hence, the equal-fragment model does not capture the observed transfer from large to small colonies (**Figure 2D** and **Appendix 1—figure 4**). The use of the erosion mechanism for the fragment distribution function is further corroborated by previous studies simulating bacterial aggregates (**Byrne et al., 2011**). For $C_1$ colonies at this intense dissipation rate, the model predicts that their median diameter remains near $l^* = 1$ (i.e. single cells). However, the experimental results show a slightly wider size distribution of both the small and large colonies than predicted by an ideal erosion mechanism (**Figure 2C**). Nevertheless, the global behavior of the size distribution is well described by the two-category model. Furthermore, a very good agreement is also observed between the measured fragmentation frequency and the model predictions given by **Equation 5** (**Figure 2F**).

To complement the mathematical model and to further investigate the mechanical origins of the resistance of division-formed colonies to intense shear, we performed rheology tests on the cyanobacterial colonies (Materials and methods). Simple and oscillatory-shear measurements revealed elasto-viscoplastic mechanics (**Balmforth et al., 2014**; **Bonn et al., 2017**) of the concentrated aggregate with a yield stress of $\tau_y = 4.3 \pm 0.3$ Pa. The low volume fraction of cells in aggregate (~12 %) indicates that the mechanical properties are mainly dictated by the EPS layer. Equating the average hydrodynamic shear stress to the yield stress in **Equation 11** and subsequently substituting the critical dissipation rate in **Equation 3** results in a critical colony size of $l_2^* = 2.8$. The eroded fragments observed in the fragmentation experiments (**Figure 2E**) are mostly smaller than this critical size. This suggests that large colonies will suffer erosion when the local hydrodynamic stress exceeds the yield stress of the EPS layer. Although the fracture behavior of colonies in suspension is expected to be different from a concentrated aggregate under shear rheology, the latter provides a reasonable and convenient estimate of the material properties of the colonies. Related to the critical size of the colony and their mechanical properties is the exponent $q_2$ (see **Equation 5**). The best fit of the results in **Figure 2** provided $q_2 = 4.1$. This value lies in the upper limit of previous results for colloidal aggregates (**Flesch et al., 1999**; **Bäbler et al., 2008**). Note that for colloids, $q$ is related to the stress concentration around cracks in the fractal structure of the aggregate (**Zaccone et al., 2009**), and low values of $q$ are associated with brittle deformation, in which the stress concentration grows quickly with aggregate size. In contrast, cyanobacterial colonies have a soft plastic behavior that is less sensitive to large cracks, thus justifying the large value of $q$.

We further tested the model with the aggregation experiments. A suspension of $C_1$ colonies with an initial distribution close to the single-cell size (**Figure 3A**) was simulated with an equal-fragment hypothesis. The model agrees well with the experimental results for the median colony size, accurately predicting a steady-state value when there is a balance between collisions and fragmentation (**Figure 3B**). The best fit of the results provided a small value for the stickiness parameter ($\alpha_1 = 0.023$). The aggregation rate is linearly dependent on the stickiness and the total biovolume fraction (**Equation 2**), thus an increase in these parameters ($\alpha_1$ and $\phi$) allows the suspension to

reach the steady-state size faster. However, the steady-state size has low sensitivity to these parameters ($\alpha_1$ and $\phi$), as the fragmentation frequency has a higher-order dependence with colony size (**Figure 2F**). The equal-fragment hypothesis leads to a single-peaked size distribution, whose steady-state median increases with the total biovolume fraction. If an ideal erosion mechanism was assumed for the $C_1$ colonies, the peak would be fixed around the single-cell size (**Appendix 1— figure 5**). The experimental results for the $C_1$ colonies are in better agreement with the equal-fragment hypothesis (**Figure 3C**).

Note that the ideal erosion and equal-fragment mechanisms are idealized scenarios, and the real distribution is expected to be more dispersed. Furthermore, a fractal description of small colonies is limited, as a continuum function between biovolume and diameter of colonies may not hold. Moreover, the turbulence model used for fragmentation (**Equation 5**) poses two limitations: (i) the theory was developed for homogeneous Gaussian turbulence (**Bäbler et al., 2008**), which differs considerably from the boundary-layer turbulence generated in a cone-and-plate shear (**Grad and Einav, 2000**), and (ii) the colonies were larger than the Kolmogorov scale for the intense dissipation rate used in our fragmentation measurements ($\eta = 20\ \mu m$ for a dissipation rate of $\dot{\varepsilon} = 5.8\ \mathrm{m^2/s^3}$, Appendix 1), meaning that the colony size is comparable to the smallest eddies in the flow. Despite these limitations, the proposed two-category model provides a good approximation of the fragmentation process observed in our experiments. Simulations that take into account spatially inhomogeneous and non-Gaussian turbulence are demanding and lack an explicit analytic expression for $\kappa(l)$ that can be applied to a population model (**Babler et al., 2015**).

## Field samples of *Microcystis* are more resistant to fragmentation due to thicker EPS layer

Samples of *Microcystis* spp. were collected from Lake Gaasperplas – the Netherlands (Materials and methods). A major difference in fragmentation behavior is observed between the laboratory cultures of *Microcystis* strain V163 and the field samples of *Microcystis* spp. While the laboratory cultures suffered intense fragmentation at a dissipation rate of $\dot{\varepsilon} = 5.8\ \mathrm{m^2/s^3}$, the colonies collected from the field showed negligible fragmentation at this dissipation rate (**Figure 4A**). The size distribution of field colonies also had a bimodal shape both initially and after 1 hr of flow (**Figure 4C and E**). The small colonies displayed an increase in size due to aggregation up to 2.5 times the initial median size (**Figure 4D**), although the fraction of small colonies remained below 11% even after a few hours of shear (**Figure 4F**). The large colonies displayed only a small increase in size due to aggregation and quickly reached a steady-state size.

The higher resistance of field colonies to shear can be attributed to the thick EPS layer that surrounds the colony, which could be seen under brightfield microscopy using a negative staining (**Figure 4B**, Materials and methods). In contrast, this layer was barely visible in colonies of laboratory cultures of *Microcystis* strain V163. Nevertheless, the limited amount of aggregation observed for large field colonies indicates that aggregation-formed colonies have a smaller resistance to shear than division-formed colonies, similar to the results observed for laboratory colonies (**Figure 3**). Differences in the morphology of colonies from field samples and laboratory cultures are frequently observed in the literature, with the latter often presenting only single cells or small colonies (**Xu et al., 2016b**). Colony formation may be influenced by various factors (**Xiao et al., 2018**), where especially high calcium and low nitrogen concentrations are identified as factors promoting high EPS production (**Gu et al., 2020**; **Chen and Lürling, 2020**; **Wang et al., 2011a**). This is consistent with the calcium and nitrogen concentrations measured in Lake Gaasperplas (71 mg/l $Ca^{2+}$ and 0.04 mg/l N) versus BG-11 medium (9.8 mg/l $Ca^{2+}$ and 246 mg/l N) (Materials and methods).

## Aggregation, fragmentation, or cell division? A guideline

Under which conditions are aggregation, fragmentation, or cell division the dominant processes in colony formation? In a quiescent fluid, cyanobacterial colonies are expected to grow solely due to cell division with a specific growth rate $\mu = N_c^{-1} dN_c/dt$, which is a function of various environmental factors such as light intensity and nutrient availability (**Huisman et al., 2004**; **Xiao et al., 2018**). Let $\bar{l} = \int_0^\infty l\, n(l,t) dl$ be the average colony size of a suspension. Assuming that the cells remain attached after division, the rate of increase in the average colony size will be:

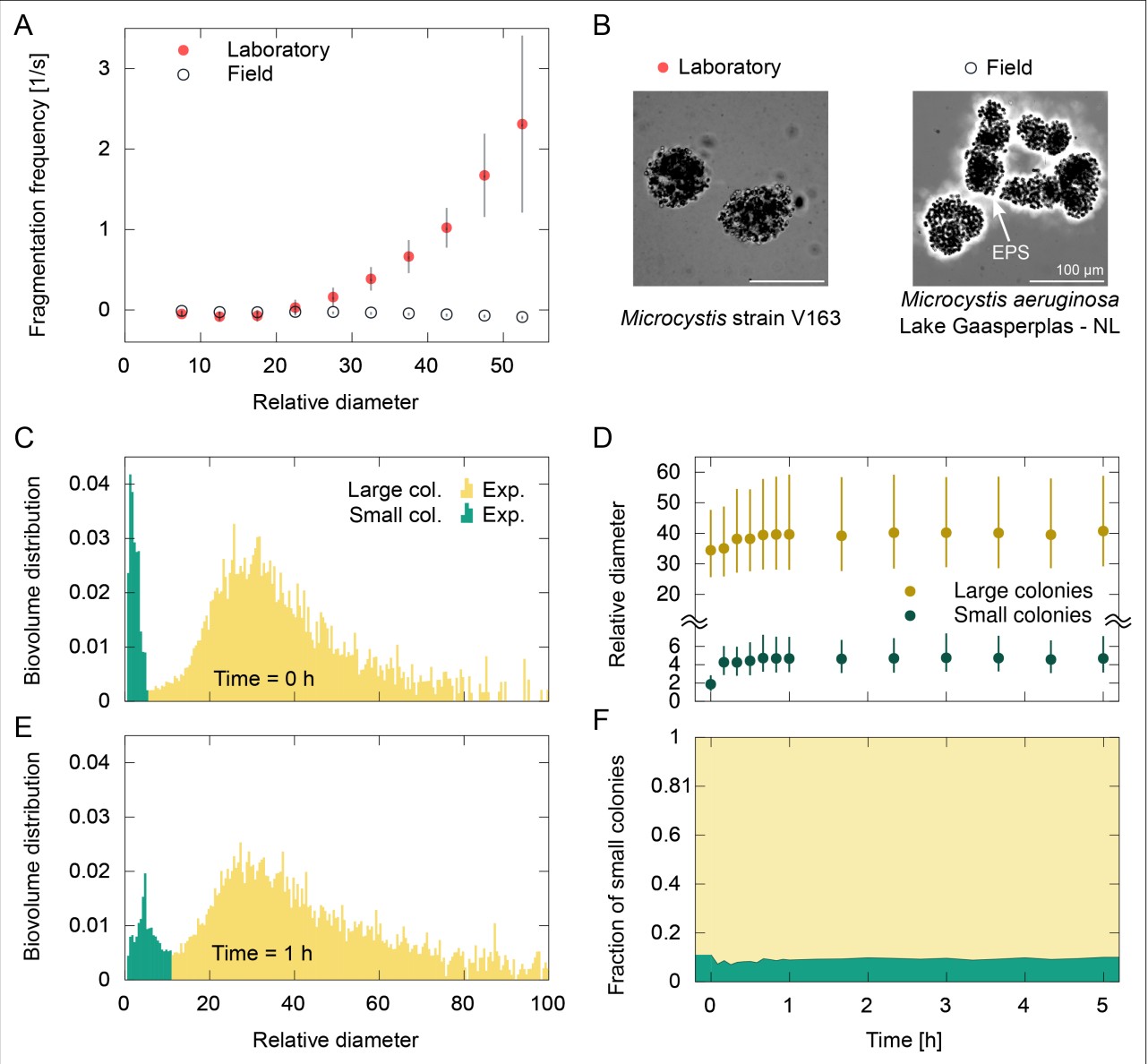

**Figure 4.** Kinetics of the fragmentation of colonies in field samples of *Microcystis* spp. at an intense dissipation rate ($\dot{\varepsilon} = 5.8\,\mathrm{m^2/s^3}$) and total biovolume fraction of $\phi = 10^{-4}$. (**A**) Comparison of the fragmentation frequency as a function of colony size for the laboratory culture (*Microcystis* strain V163) and the field samples (*Microcystis* spp.). Bars indicate the uncertainty in the fragmentation frequency propagated from the uncertainty of the concentration distribution (Appendix 1). (**B**) Brightfield microscopy images of colonies in a Nigrosin-dyed medium (dark region) show evidence of a thick extracellular polymeric substance (EPS) layer (bright region) surrounding a field colony. (**C**) Initial size distribution of colonies in field samples as a function of the relative diameter. The size distribution had a bimodal shape, with small colonies (green) and large colonies (yellow). (**D**) The median diameter of the colonies in each subpopulation as a function of time. Bars indicate 25th and 75th percentiles. (**E**) After 1 hr of shear flow, the small colonies aggregated slightly, while the large colonies kept their size distribution. (**F**) The biovolume fraction of small colonies remained nearly constant during the experiment.

$$\frac{1}{\bar{l}}\frac{\mathrm{d}\bar{l}}{\mathrm{d}t} = \frac{\mu}{f}\,. \tag{7}$$

The increase in colony size will progress until a maximum size $l_m$ is reached. The upper limit in colony size is still a subject of intense discussion, with species variability, reduced division rate at large sizes, and dissolution of the EPS layer being important factors (*Xiao et al., 2018*; *Feng et al., 2020*). If the suspension is concentrated enough, collisions between colonies will also increase the average colony size due to aggregation. Let us consider initially a monodisperse suspension with a single size

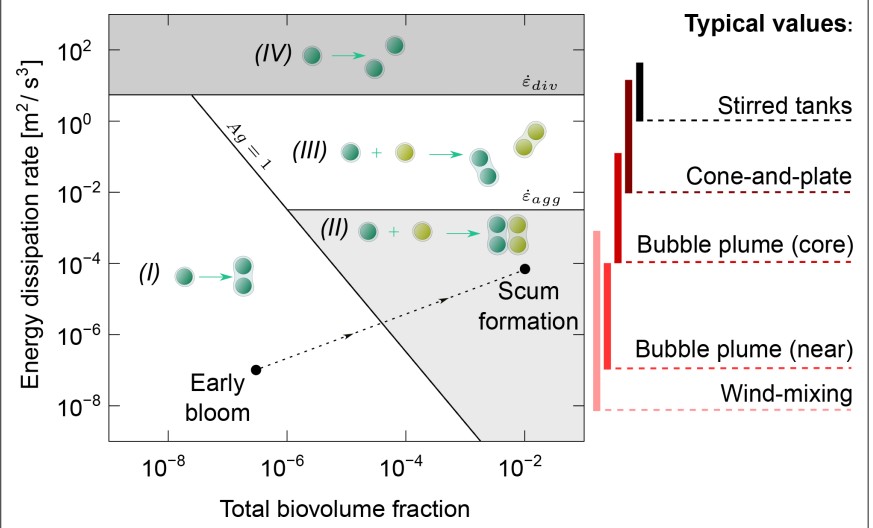

**Figure 5.** Phase diagram indicating the dominant colony formation mechanism as a function of the dissipation rate $\dot{\varepsilon}$ and the cyanobacterial abundance (expressed by the total biovolume fraction $\phi$). (I) Colonies grow only by cell division at low dissipation rates and total biovolume fractions. (II) As the biovolume fraction increases, aggregation enhances colony growth. (III) For moderate dissipation rates, aggregated colonies are fragmented, and only cell division can increase colony size. (IV) Fragmentation of colonies dominates at intense dissipation rates, irrespective of whether these colonies were formed by aggregation or cell division. Bars on the right side indicate typical values of dissipation rate observed for natural wind-mixing, bubble plumes in artificially mixed lakes, and laboratory-scale setups such as cone-and-plate systems and stirred tanks. The dashed arrowed line indicates the transition from an early bloom (left bullet – WHO alert level 1, ***Chorus and Welker, 2021***) to a dense surface scum (right bullet – typical scum biovolume fraction, ***Wu et al., 2020***) under typical wind mixing.

$\bar{l}$. At short times, the rate of increase in the average colony size due to aggregation will be (Appendix 1):

$$\frac{1}{\bar{l}}\frac{\mathrm{d}\bar{l}}{\mathrm{d}t} = 2.49\,(2^{1/f} - 1)\,\phi\,\alpha\,\left(\bar{l}\right)^{4-2f}\sqrt{\frac{\dot{\varepsilon}}{\nu}}\,. \tag{8}$$

Assuming $f = 2$ for simplicity (which is similar to the measured value for *Microcystis* strain V163), we can use ***Equations 7 and 8*** to define a dimensionless number $Ag$, quantifying the ratio between the rates of increase in the average colony size due to aggregation and cell division, given by:

$$Ag = 2.06\,\frac{\alpha\,\phi}{\mu}\,\sqrt{\frac{\dot{\varepsilon}}{\nu}} \tag{9}$$

The dimensionless ratio $Ag$ reveals the dominant mechanism for colony formation (cell division is dominant for $Ag \ll 1$, while aggregation is dominant for $Ag \gg 1$) and depends on the total biovolume fraction $\phi$ and dissipation rate $\dot{\varepsilon}$, as shown by the phase diagram in ***Figure 5***. For a lake with a total biovolume fraction corresponding to an early bloom ($\phi = 3 \cdot 10^{-7}$, according to WHO alert level 1, ***Chorus and Welker, 2021***), a low dissipation rate typical of wind-induced turbulence ($\dot{\varepsilon} \sim 10^{-7}\,\mathrm{m^2/s^3}$ at a depth of $\sim 1\mathrm{m}$, ***Wüest and Lorke, 2003***), a specific growth rate of $\mu \sim 0.008\,\mathrm{hr^{-1}}$(***Huisman et al., 2004***), a kinematic viscosity of $\nu = 10^{-6}\,\mathrm{m^2/s}$ and a stickiness of $\alpha \sim 0.02$, we estimate a ratio $Ag \sim 10^{-3}$. Under these conditions, the colony size increases due to cell division while the role of aggregation is negligible (region I in ***Figure 5***). In contrast, during a dense bloom event, the scum layer can reach a very high biovolume fraction ($\phi \sim 10^{-2}$, ***Wu et al., 2019***; ***Wu et al., 2020***) in the top layer (depth of $\sim 10^{-2}\,\mathrm{m}$), where concomitantly the dissipation rate is much higher ($\dot{\varepsilon} \sim 10^{-4}\,\mathrm{m^2/s^3}$ under weak wind turbulence, ***Wang and Liao, 2016***), which leads to $Ag \sim 10^{5}$ (keeping all remaining parameters constant). Although this regime (region II in ***Figure 5***) was not explored in our cone-and-plate experiments, recent mesocosm experiments found clear evidence for the aggregation of *Microcystis* colonies in surface scums, with rapid increase in colony size at rates exceeding cell division rates (***Wu et al., 2024***). Specifically, for a bulk biovolume fraction of $\phi \sim 10^{-4}$ and a dissipation rate of

$\dot{\varepsilon} \sim 10^{-4} \, \text{m}^2/\text{s}^3$, that study measured a rate of increase in the average colony size of $\bar{l}^{-1} \text{d}\bar{l}/\text{d}t \sim 0.2 \, \text{hr}^{-1}$, corresponding to a ratio of $Ag \sim 10^1$. These results lie well inside region II of *Figure 5*. These mesocosm experiments confirm our model prediction that aggregation is the dominant mechanism for colony formation in region II.

Fragmentation limits the maximum colony size to a critical value $l^*$. Our experimental results (*Figure 5*) indicate that aggregated colonies have a lower critical size than division-formed colonies for the same turbulence intensity. Conversely, a higher critical dissipation rate is needed for division-formed colonies ($\dot{\varepsilon}_{div}$) than for aggregated colonies ($\dot{\varepsilon}_{agg}$) to achieve the same critical size $l^* = l_m$. This creates an interesting regime at high biovolume fractions and dissipation rates in the range $\dot{\varepsilon}_{agg} < \dot{\varepsilon} < \dot{\varepsilon}_{div}$, in which only division-formed colonies can survive the turbulent flow, even though aggregation is faster than division ($Ag \gg 1$) (region III in *Figure 5*). Typical values of critical dissipation rate for our experiments are $\dot{\varepsilon}_{agg} = 2.1 \cdot 10^{-3} \, \text{m}^2/\text{s}^3$ and $\dot{\varepsilon}_{div} = 3.7 \, \text{m}^2/\text{s}^3$, assuming a maximum colony size of $l_m = 50$. Under natural turbulent conditions (e.g. surface wind-mixing), the dissipation rate ranges from $\dot{\varepsilon} \sim 10^{-8} \, \text{m}^2/\text{s}^3$ (*Wüest and Lorke, 2003*) to peaks of $\dot{\varepsilon} \sim 10^{-3} \, \text{m}^2/\text{s}^3$ at strong winds (in the absence of wave breaking), which is still insufficient to induce colony fragmentation (*Wang and Liao, 2016*). Artificial mixing systems can achieve levels of turbulence that are many orders of magnitude higher. In the core of bubble plumes, dissipation rates of $\dot{\varepsilon} \sim 10^{-4} - 10^{-1} \, \text{m}^2/\text{s}^3$ are reached (*Lai and Socolofsky, 2019*). This value was sufficient to prevent aggregation of large colonies in our experiments but was not high enough to induce fragmentation of division-formed colonies. Moreover, colonies remain only a short time (~1 min) in the bubble plume core, after which they are dispersed to weaker turbulent regions (dissipation rates of $\dot{\varepsilon} \sim 10^{-7} - 10^{-4} \, \text{m}^2/\text{s}^3$ just a few meters away from the bubble plume core). Nevertheless, bubble plumes can prevent the formation of a scum layer not only by fragmentation of newly formed weak aggregates but also by providing sufficient turbulent mixing to disperse the buoyant colonies, thereby reducing the local biovolume fraction at the surface (*Visser et al., 2016a*). The intense hydrodynamic stress required to fragment division-formed colonies (region IV in *Figure 5*) can only be achieved by mixers with fast-moving surfaces, such as the cone-and-plate setup used in our experiments ($\dot{\varepsilon} \sim 10^{-2} - 10^1 \, \text{m}^2/\text{s}^3$) or propellers used in stirred tanks ($\dot{\varepsilon} \sim 10^0 - 10^2 \, \text{m}^2/\text{s}^3$, *Laufhütte and Mersmann, 1987*; *Peter et al., 2006*). These mixing devices can provide intense dissipation rates in small fluid volumes, but their implementation across entire lakes would be excessively energy-consuming.

## Discussion

Despite the widespread presence of bacterial colonies across species and environmental habitats, the influence of fluid flows on colony formation remains a fertile ground for continued exploration. A significant challenge is the lack of a generic framework in which the aggregation and fragmentation of colonies can be studied under various hydrodynamic stresses. We take a major step toward overcoming this challenge by presenting a combination of experiments and theory to elucidate the underlying mechanisms that determine the size distribution of cyanobacterial colonies over a wide range of flow intensities. Our experiments show that division-formed *Microcystis* colonies can be fragmented under intense dissipation rates $\dot{\varepsilon} \sim 3 - 10 \, \text{m}^2/\text{s}^3$, following an erosion mechanism. Previous studies *O'Brien et al., 2004*; *Li et al., 2018* have observed significant fragmentation of *Microcystis* colonies in field samples at dissipation rates as low as $\dot{\varepsilon} \sim 10^{-4} \, \text{m}^2/\text{s}^3$. However, such studies primarily report volume averages of the dissipation rate, overlooking the likelihood of fragmentation occurring near fast-moving solid surfaces (grid stirrer, *O'Brien et al., 2004*, and propeller, *Li et al., 2018*), where the local hydrodynamic stresses are orders of magnitude higher. Instead, our results indicate that fragmentation of division-formed colonies cannot be achieved with wind mixing or aeration systems.

We have demonstrated that flow-induced aggregated colonies of *Microcystis* form weaker structures than division-formed colonies, likely due to the weak EPS bonds between cells. A previous study on colony aggregation at high $Ca^{2+}$ levels observed similar morphological differences in colony formation (*Chen and Lürling, 2020*). There, an initial fast cell aggregation produced a sparse colony structure, followed by a more compact structure of the colonies associated with cell division. Furthermore, we showed that colony aggregation is unlikely in lakes under early bloom conditions, as the collision frequency of colonies is too low. Our results, based on the mechanical properties of colonies, are corroborated by previous studies based on the sequencing of individual *Microcystis* colonies

(*Pérez-Carrascal et al., 2021*; *Smith et al., 2021*). These studies have identified that the genetic diversity within *Microcystis* colonies is much smaller than the diversity between *Microcystis* colonies, thus supporting the hypothesis that colony formation is dominated by cell division rather than by aggregation of colonies. Nonetheless, our model supports recent studies which demonstrated that, during scum formation in very dense blooms or phytoplankton patches, flow-induced aggregation likely plays a role and may quickly increase colony size (*Wu et al., 2019*; *Wu et al., 2020*). A summary of these outcomes is presented in the phase map illustrated in *Figure 5*.

From a broader perspective, the technological advance of combining microscopic imaging with rheometry has relevance beyond predicting and mitigating cyanobacterial blooms. Since fluid flow affects the colonies and aggregates of many species (*Burd and Jackson, 2009*; *Drescher et al., 2013*; *Rusconi et al., 2014b*; *Goldstein, 2015*; *Thomen et al., 2017*; *Ros-Rocher et al., 2026*; *Song and Rau, 2022*), the experimental methods and theoretical framework presented here, along with the identified mechanisms, could serve as a foundation to better understand the relationship between fluid mechanics and biological aggregates, while keeping a keen eye on the wide variation of biological functions and morphodynamics present in biological aggregates exposed to hydrodynamic stresses. An interesting example is a recent study on the fragmentation of marine snow consisting of diatoms and microplastics, which used similar methods as in our study (*Song and Rau, 2022*; *Song et al., 2023*). In contrast to the high mechanical resistance of *Microcystis* colonies, these marine snow aggregates displayed a low critical dissipation rate. Furthermore, their fragmentation was not mediated by an erosion mechanism, but by neck formation (i.e. deformation is higher in a small cross-sectional area in the middle of the aggregate). Other examples include non-natural settings, such as algae or cyanobacteria cultured in bioreactors, where dense cell aggregates experience intense mixing (*Ducat et al., 2011*; *Johnson et al., 2018*). The methods and results presented here could serve as design guidelines for the study of these applications.

## Materials and methods
### Sample preparation

*Microcystis* strain V163 cultured under laboratory conditions, as well as samples collected at Lake Gaasperplas – The Netherlands containing *Microcystis* spp. colonies were used for the experiments (*Figure 1*). *Microcystis* strain V163, originally isolated from Lake Volkerak – The Netherlands (*Kardinaal et al., 2007*), grows in colonies and was obtained from the algal collection of the Institute for Biodiversity and Ecosystem Dynamics, University of Amsterdam. The *Microcystis* strain V163 was batch-cultured in Erlenmeyer flasks with standard BG-11 medium. The cultures were kept in an incubation chamber at a temperature of 20°C. The light intensity was kept at 12 $\mu$mol photons m$^{-2}$ s$^{-1}$ with a light:dark cycle of 14 hr:10 hr. The cultures were diluted with new BG-11 medium every week and kept at a maximum total biovolume fraction of $\phi = 5 \cdot 10^{-4}$. The cultures were not shaken except after the addition of medium and before filtration. The *Microcystis* strain V163 has a single-cell diameter of $d_1 = 6.8 \pm 1.4$ μm (SD, *N*=189). We categorize partially divided cells also as single cells, as these also behave as single units in a fragmentation/aggregation process. When considering only spherical cells not undergoing division, a diameter of $5.1 \pm 0.8$ μm (SD, *N*=328) was obtained. Colonies of up to 500 μm were present in the culture under the incubation conditions as described before.

Samples of *Microcystis* spp. were collected from Lake Gaasperplas – the Netherlands (Appendix 1 for water monitoring data, location of sampling point) on August 14, 2022. Water from the surface layer was filtered over a plankton net with a 100 μm mesh size. Samples were kept refrigerated at 4°C, and the experiments were conducted over the next 3 days. The plankton composition was dominated by *Microcystis* spp. ($89 \pm 7\%$, SD, $N = 231$), with *M. aeruginosa* being the main morphospecies (*Appendix 1—figure 6*), with an average single-cell diameter of $d_1 = 7.0 \pm 1.4$ μm (SD, $N = 100$). Colonies of *Dolichospermum* spp. were present in lower numbers. However, no differentiation of colony species was made during image processing, as it was not feasible to isolate them.

Negative staining was used to verify the presence of an EPS layer surrounding the colonies of *Microcystis* strain V163 and of *Microcystis* spp. in field samples. Nigrosin was added to the surrounding medium, causing the EPS layer to appear bright under brightfield microscopy.

Upon appropriate filtering of the culture, we obtained a suspension with a specific size range. Filter paper with a 25 μm pore size was used to obtain a suspension primarily consisting of single

cells (colonies with 2 or 3 cells also present in small numbers). For a suspension mainly comprising of large colonies, we filtered the culture with a 53 μm nylon mesh. The colonies collected in the mesh were then redispersed in a new BG-11 medium. To estimate the total biovolume fraction of the suspension, 1 ml of the sample underwent a heat treatment to disaggregate colonies (**Humphries and Widjaja, 1979**). At least 1000 cells were counted in a Neubauer hemocytometer assisted by the image processing software ImageJ. The total biovolume fraction of the suspension was calculated as $\phi = cv_1$, where $c$ is the cell concentration and $v_1$ is the average single-cell volume. Prior to each experiment, the desired biovolume fraction was adjusted using new BG-11 medium for the *Microcystis* strain V163 culture and filtered (0.8 μm) water from Lake Gaasperplas for field samples of *Microcystis* spp.

## Cone-and-plate shear

A cone-and-plate shear flow was selected for our experiments (**Figure 1B** and **Appendix 1—figure 7**). This geometry is commonly used for standard rheological measurements because it provides a homogeneous shear rate under a laminar flow regime. Although less common in inertial regimes, it has been recommended by previous studies to investigate the effects of hydrodynamic stress in tissues (**Grad and Einav, 2000**; **Buschmann et al., 2005**; **Ye et al., 2019**). Furthermore, the mixing provided by turbulence helps maintain a homogeneous concentration throughout the suspension.

Our setup consists of an upper conical probe made from transparent acrylic, with an outer diameter of 42 mm and a cone complementary angle of $\theta$=5.44° (**Appendix 1—figure 7A**). The conical probe is attached to a steel shaft, and it is driven by a rheometer head Anton Paar DSR 502. The lower surface consists of a 1-mm-thick glass slide. A cylindrical acrylic ring confines the fluid, preventing spillage of the sample due to strong centrifugal forces. The inner diameter of the ring is 46 mm, which ensures that the nominal shear rate between the upper probe's outer surface and the ring is the same as between the conical and lower surfaces. A clearance gap of 10 μm was set between the tip of the cone and the lower surface to avoid solid-solid friction.

At low rotational speeds, the flow is laminar with a homogeneous shear rate $\dot{\gamma}_n = \omega / \theta$, where $\omega$ is the angular velocity of the conical probe. The flow transitions to a turbulent regime at faster rotations, as the inertial forces become dominant. Since the shear rate is not uniform at the nonlinear regime, the nominal shear rate $\dot{\gamma}_n$ is no longer an appropriate measure for shear intensity. Instead, the average intensity is given by the energy dissipation rate $\dot{\varepsilon}$, which measures the rate of energy loss by viscous dissipation per unit fluid mass. Under steady-state conditions, this dissipation rate balances the power injected into the fluid, therefore:

$$\dot{\varepsilon} = \frac{T\omega}{\rho V}, \tag{10}$$

where $\rho$ and $V$ are, respectively, the density and volume of the fluid sample, and $T$ is the torque on the conical probe, measured by the rheometer head. The flow regime can be identified from the energy dissipation rate curve against the angular velocity, with $a \sim \omega^2$ dependence in the laminar regime and $a \sim \omega^{5/2}$ dependence in the turbulent regime (**Appendix 1—figure 7B**). All shear experiments conducted in this work were in the latter regime. The average hydrodynamic shear stress can be calculated from the dissipation rate by:

$$\overline{\tau}_t = \rho \sqrt{\nu \dot{\varepsilon}}. \tag{11}$$

Our cone-and-plate setup covers energy dissipation rates over three orders of magnitude, from $10^{-2}$ to $10 \, \mathrm{m^2/s^3}$. At the lower end, the dissipation rate is limited by the sedimentation of colonies near the stagnation point at the conical tip. Due to the small gap, this stagnation point is not accessible to the colonies. A large colony of $l \sim 50$ does not fit within the surfaces up to a radial distance from the tip of 4 mm. For the lowest dissipation rate ($\dot{\varepsilon} = 0.019 \, \mathrm{m^2/s^3}$), a minimum secondary flow speed of $\sim 100 \, \mu\mathrm{m/s}$ is present (**Sdougos et al., 1984**). This value is above the sedimentation speed of colonies of *Microcystis* strain V163, which are either buoyant or slightly sinking due to the presence of gas vesicles. Furthermore, the region near the conical tip accounts for a small fraction of sample volume, while the velocity fluctuations in the bulk region of the cone-and-plate setup ensure that the colonies remain in suspension and are homogeneously distributed. The upper limit for the operating dissipation rate is set by the spillage of fluid out of the cylindrical confinement. Regarding the optimal range of biovolume fraction, this is limited at $\phi = 0.0005$, above which the optical overlap between nearby

colonies influences the colony count. The kinematic viscosity of the liquid $v$ can be taken as equal to that of water for the biovolume fraction explored here.

The experimental setup provides a configuration where the dissipation rate can be controlled over a large range, while stress values can be measured with high precision. This, combined with optical access and simultaneous microscopy, presents a well-suited tool to study the response of biological samples under fluid flow. Our setup configuration was optimized to cover the dissipation rates observed for deep artificial mixing systems. However, the conical probe diameter could be increased, and the rotation speed reduced simultaneously to cover a lower range of dissipation rate typical of wind mixing. Furthermore, the gap between the surfaces can be easily adjusted to fit other organisms.

## Fragmentation and aggregation experiments

The fragmentation and aggregation experiments involved subjecting a suspension of *Microcystis* colonies to the cone-and-plate shear and using brightfield microscopy to image the colonies under shear. The lower surface was positioned on top of an inverted microscope Nikon TS100 with a 4× objective lens. The center of the field of view was located 15 mm from the rotation axis and with the focal plane set 0.5 mm above the lower surface. Illumination was provided by a Schott KL 1500 HAL light source equipped with a cyan filter (low pass with a cutoff wavelength of 555 nm) to minimize the photosynthetic activity of the sample. Images were captured with a Prime BSI Express sCMOS camera with a resolution of 1.63 μm per pixel. Prior to each experiment, a reference background image was created using distilled water, which was then discarded. A volume of 3.5 ml of the suspension of *Microcystis* colonies was pipetted in between the shear surfaces. The shear protocol was as follows: pre-shear at $\dot{\varepsilon} = 1.1 \, \text{m}^2/\text{s}^3$ for 1 min to mix the suspension; 12 shear cycles of 5 min each, followed by 12 cycles of 20 min each at the target dissipation rate. Imaging cycles, at moderate dissipation rate of $0.019 \, \text{m}^2/\text{s}^3$ for 1 min, were intercalated with each shear cycle. During the imaging cycles, 200 frames were captured at a rate of 10 fps, ensuring that each frame was sampled over a different set of colonies. Examples of the captured frames are shown in *Figure 1*.

## Image analysis

The frames underwent pre-processing with background subtraction and thresholding (Appendix 1). Image segmentation was performed with the *Scikit Image* library for Python (*van der Walt et al., 2014*). Colony size was determined by their Feret diameter $d$, which represents the maximum distance between two boundary points of the colony. The number of cells in a colony was estimated from the diameter using the formula $N_c = (d/d_1)^f$, where $f$ is the average fractal dimension of the colonies. Since not all colonies were within the depth-of-focus, a correction function for the colony count was estimated using a suspension with a known size distribution. The image magnification was optimized for intermediate and large colonies, given the wide range of colonies (from single cells ≈7 μm to large colonies ≈500 μm). Colonies smaller than 25 μm were measured with lower precision, necessitating a second correction function for accurate counting (Appendix 1 – *Equation 13*). Typical uncertainties for the biovolume distribution are displayed in *Appendix 1—figure 8*. The uncertainty is higher for small colonies due to the uncertainty in image resolution, and also higher for very large colonies due to the low counts.

To estimate the colony fractal dimension $f$, a reference sample of colonies had its size distribution characterized in a cylindrical counting chamber, followed by the disaggregation into single cells and the measurement of cell concentration in a hemocytometer. The fractal dimension was computed by fitting the equation, $cV = \sum_k l_k^f$, where $c$ is the cell concentration and $l_k$ is the relative diameter of each colony in the measured volume $V$. For *Microcystis* strain V163, a value of $f = 2.09$ was obtained, while for the field samples of *Microcystis* spp. it was 1.64. At least 3000 colonies were counted for each sample, giving a maximum uncertainty of $\delta f < 0.04$.

For the direct computation of the fragmentation frequency, we assume a pure fragmentation with an ideal binary erosion, such that the time derivative of the colony size distribution is:

$$\frac{\mathrm{d}}{\mathrm{d}t} N[i](t) = -\frac{\kappa[i] \, N[i](t)}{l^f[i] - l^f[i-1]} + \frac{\kappa[i+1] \, N[i+1](t)}{l^f[i+1] - l^f[i]} \,, \tag{12}$$

where $N[i](t)$, $\kappa[i]$, and $l[i]$ refer to the concentration distribution (number of counted colonies per volume per bin width), fragmentation frequency, and relative diameter at the center of bin $i$. This

formula is only valid for the bins larger than the single cell, as the latter does not suffer fragmentation. The concentration distribution $N[i]$ is initialized with the experimental value for $t = 0$, and an initial guess for the fragmentation frequency $\kappa[i]$ is made for all bins larger than a single cell. The values of $N[i](t)$ are evolved in time using an implicit Radau scheme. For each bin, the cumulative squared error between the experimental and numerical values of $N[i](t)$ are computed, and the error is minimized using the values of $\kappa[i]$ as the optimization parameters. The uncertainty of $\kappa[i]$ is propagated from the uncertainty of the concentration distribution. The script for image analysis is available at https://github.com/FluidLab/CyanoB_Agg_Frag (copy archived at *Sinzato, 2026*). Detected features for all datasets and raw images for the dataset in *Figure 2A–E* are available at https://doi.org/10.5281/zenodo.11098426.

## Yield stress of cyanobacterial colonies

The mechanical resistance of *Microcystis* colonies can be estimated by separating the colonies from the medium and measuring the yield stress of the concentrated aggregate. A suspension of *Microcystis* strain V163 was filtered through a 53 μm mesh, and the large colonies were resuspended in fresh BG-11 medium. The sample was centrifuged for 30 min at 1000×$g$, and the pellet was separated. A dense gelatinous phase composed of cells and EPS was obtained, with cells accounting for 12% of the total biovolume fraction. The yield stress of the concentrated aggregate was measured with an Anton Paar MCR 301 rheometer. The upper probe was a sand-blasted 50 mm cone with a 1° angle, while the lower surface was covered with sandpaper to avoid wall slip. Each sample was pre-sheared at 10 s$^{-1}$ for 2 min. Steady-shear and large amplitude oscillatory-shear tests were performed (*Appendix 1—figure 9*). From a Herschel-Bulkley fit over the steady-shear data, $\tau = \tau_y + k\dot{\gamma}^n$, the dynamical yield stress $\tau_y = 4.3 \pm 0.3$ Pa (SD from best fit) was estimated. Due to the limited volume of samples available, only one repetition was possible for steady-shear and oscillatory-shear tests each.

## Simulations

The balance *Equation 1* for the total normalized biovolume distribution was solved numerically using a zero-order discrete population balance, or sectional method, in which the size distribution is described by uniformly sized bins of width $\Delta l = 0.5$, and the biovolume distribution is evaluated at a representative size centered on the bin (*Vanni, 2000*). The integrals for the aggregation (*Equation 2*), fragmentation (*Equation 4*), and transfer (*Equation 6*) rates were discretized with the approach by *Kumar and Ramkrishna, 1996*. The biovolume distribution evolved in time with a Runge-Kutta 4–5 adaptive time step scheme written in Fortran. The simulation results were fitted with the experimental data by minimizing the sum of absolute errors of each category's 25th, 50th, and 75th percentiles of the biovolume distribution, using a Nelder-Mead algorithm (*Gao and Han, 2012*). The stickiness $\alpha_1$ and for the pair of constitutive parameters $S_1$ and $q_1$ for the critical size of small colonies (category $C_1$) were obtained by fitting the model to the dataset of a single-cell suspension under $\dot{\varepsilon} = 0.019$ m$^2$/s$^3$ and $\phi = 10^{-4}$. The pair of constitutive parameters $S_2$ and $q_2$ for the critical size of large colonies (category $C_2$) were obtained by fitting the model to the dataset of a large colony suspension under $\dot{\varepsilon} = 5.8$ and $\phi = 10^{-4}$. These datasets were chosen as their aggregation/fragmentation times were similar to the measurement time, while for the other datasets, the variations were either too fast or slow to be captured in detail. In our simulations, the stickiness parameter $\alpha_1$ controls the rate at which an initial single-cell suspension achieves an aggregated steady state. The parameters $S_i$ are the dissipation rates for which all colonies in category $i$ are broken, and a higher $S_i$ shifts the steady-state colony size upward. Finally, the exponents $q_i$ controls the sensitive of the fragmentation frequency to colony size. A high value of $q_i$, as observed in our experiments, implies a low sensitivity to size and leads to a broader steady-state size distribution. The script for the simulation is available at https://github.com/FluidLab/CyanoB_Agg_Frag copy archived at *Sinzato, 2026*.

## Acknowledgements

We thank Maria van Herk, Pieter Slot, and Merijn Schuurmans for their support with laboratory cultures. We are grateful to Axel Gunderson for his assistance with field work, and to Daan Giesen for his support with experimental setup design. We especially thank Johan Oosterbaan and Richard Steel for their contributions during the project's conception.

# Additional information

## Competing interests
Robert Uittenbogaard: Owner of Hydro-Key Ltd. The other authors declare that no competing interests exist.

## Funding

| Funder | Grant reference number | Author |
|---|---|---|
| Hoogheemraadschap van Rijnland | | Maziyar Jalaal |
| European Research Council | 2023-StG-101117025 | Maziyar Jalaal |

The funders had no role in study design, data collection and interpretation, or the decision to submit the work for publication.

## Author contributions
Yuri Z Sinzato, Conceptualization, Software, Validation, Investigation, Visualization, Methodology, Writing – original draft; Robert Uittenbogaard, Conceptualization, Methodology, Writing – review and editing; Petra M Visser, Jef Huisman, Conceptualization, Supervision, Methodology, Writing – review and editing; Maziyar Jalaal, Conceptualization, Resources, Supervision, Funding acquisition, Visualization, Methodology, Writing – review and editing

## Author ORCIDs
Yuri Z Sinzato ⓘ https://orcid.org/0000-0002-3324-5712
Petra M Visser ⓘ https://orcid.org/0000-0003-3294-1908
Jef Huisman ⓘ https://orcid.org/0000-0001-9598-3211
Maziyar Jalaal ⓘ https://orcid.org/0000-0002-5654-8505

Reviewer #2 (Public review): https://doi.org/10.7554/eLife.103503.4.sa1
Author response https://doi.org/10.7554/eLife.103503.4.sa2

---

# Additional files

## Supplementary files
MDAR checklist

## Data availability
The scripts for image analysis and model presented in this paper are openly available at https://github.com/FluidLab/CyanoB_Agg_Frag (copy archived at *Sinzato, 2026*). Detected features for all datasets and source data images for the dataset in *Figure 2A-E* have been deposited at https://doi.org/10.5281/zenodo.11098426.

The following previously published dataset was used:

| Author(s) | Year | Dataset title | Dataset URL | Database and Identifier |
|---|---|---|---|---|
| Sinzato Y | 2024 | Fragmentation and aggregation of cyanobacterial colonies under a cone and plate shear | https://doi.org/10.5281/zenodo.11098426 | Zenodo, 10.5281/zenodo.11098426 |

---

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

## Appendix 1

### Field sample collection

Samples of *Microcystis* spp. were collected from Lake Gaasperplas – the Netherlands – at the sampling point 52°18′31.0"N 4°59′59.2"E on August 14, 2022, at 14:00. This is an artificial lake used for recreational purposes. No visible surface scums were present at the time of sampling. Water quality measurements of Lake Gaasperplas were obtained from the Dutch Water Authority (Rijkswaterstaat) (https://www.waterkwaliteitsportaal.nl/oppervlaktewaterkwaliteit). *Appendix 1— figure 6* shows micrographic images of the phytoplankton samples.

### Image analysis

This section describes the routine used to analyze raw images of the aggregation and fragmentation process of *Microcystis* colonies under a cone-and-plate shear setup. Each frame has dimensions of 2048×2048 pixels. Prior to each experiment, 3.5 ml of distilled water was pipetted in the setup, and 200 frames were captured at $\dot{\varepsilon} = 0.019 \, \mathrm{m^2/s^3}$ and 10 fps. A background image was generated by averaging over the frames and applying a Gaussian blur filter (radius = 5 px). After that, the distilled water was discarded, and 3.5 ml of cyanobacteria suspension was pipetted into the setup. Each time point in the datasets corresponds to a stack of 200 frames. The features in each frame were detected and measured using two separate routines according to the size range:

- Small colonies ($l < 5$): The frame is cropped into a smaller section (400×400 pixels), to speed the processing. The background is subtracted using the reference image. The image intensity is renormalized and then thresholded. Small features ($0.5 < l < 5$) are segmented, measured, and counted.
- Large colonies ($l > 5$): A Gaussian blur (radius = 5 px) is applied to filter out small features. The background is subtracted using the reference image. The image intensity is renormalized and then thresholded. Large features ($l > 5$) are segmented, measured, and counted.

For each time point, the counted features of the 200 frames were distributed in bins of width $\Delta l = 0.5$. Since not all colonies are within the depth-of-focus, a correction function for the featured count was estimated using a suspension with a known size distribution, given by:

$$N[i] = \begin{cases} (\lambda_3 + \lambda_1 (\lambda_2 - \lambda_3)) N_p[i] & , \text{if } l[i] \leq 2 \\ (\lambda_3 + \lambda_1 (\lambda_2 - \lambda_3)(4 - l[i])/2) N_p[i] & , \text{if } 2 < l[i] \leq 4 \\ \lambda_3 N_p[i] & , \text{if } 4 < l[i] \leq 5 \\ \lambda_4 N_p[i] & , \text{if } l[i] > 5 \end{cases} \tag{13}$$

Here, $N_p[i]$ and $N[i]$ are the concentration distributions before and after correction, respectively, i.e., the number of colonies per volume per bin width in a bin centered at $l[i]$. After that, the concentration distribution is converted to biovolume distribution, $n^*[i] = N[i] \, l[i]^f$. Finally, the biovolume distribution is normalized, $n[i] = n^*[i]/\sum_j n^*[j]$.

To obtain the calibration parameters $\lambda_i$, a sample of filtered colonies of *Microcystis* strain V163 was subjected to the cone-and-plate shear setup for 2 min at $\dot{\varepsilon} = 0.019 \, \mathrm{m^2/s^3}$, and the concentration distribution before correction was measured (*Appendix 1—figure 8A and B*). Next, the upper conical surface was removed and the colonies were allowed to sediment in the lower plate. The reference concentration distribution was measured with an inverted microscope (*Appendix 1—figure 8C and D*). The parameters $\lambda_2 = 2.79$ and $\lambda_4 = 1.24$ were estimated from the ratio between the reference and original concentration distributions (*Appendix 1—figure 8E and F*) for small ($l < 5$) and large ($l > 5$) colonies, respectively. The parameter $\lambda_3 = 0.52$ ensures that the correction function is continuous between the two size ranges. *Appendix 1—figure 8G and H* displays the normalized biovolume distribution measured in the cone-and-plate shear setup after correction function. The colony counts for $l \leq 4$ have low precision due to the pixel resolution; therefore, the parameter $\lambda_1$ was introduced to ensure biovolume conservation. The value of $\lambda_1 = 1$ was used for $t = 0$, and it was estimated for each subsequent time point so that the total biovolume remained constant through time.

The relative uncertainty of the concentration distribution (error bars in *Appendix 1— figure 8A and B*) is computed from the bin counting uncertainty and the size measurement

uncertainty, $\delta N[i] / N[i] = \sqrt{(\delta_{count}[i])^2 + (\delta_{size}[i])^2}$. The bin counting uncertainty is estimated as $\delta_{count}[i] = 1/\sqrt{\text{\#colonies in bin } i}$ and the size measurement uncertainty is computed as $\delta_{size}[i] = (\Delta N[i] / N[i] \, \Delta l) \, \delta l / l[i]$, where $\Delta N[i]$ is the absolute difference between neighboring bins. The script for image analysis is available at https://github.com/FluidLab/CyanoB_Agg_Frag (copy archived at *Sinzato, 2026*).

## Fragmentation measurements with *Microcystis* strain V163

Additional measurements of fragmentation of large colonies of *Microcystis* strain V163 were performed. Colonies grown by cell division were filtered through a mesh of 53 µm, and the collected colonies were redispersed in BG-11 medium. Under a moderate dissipation rate of $\dot{\varepsilon} = 0.019 \, \text{m}^2/\text{s}^3$, the fragmentation was negligible (*Appendix 1—figure 3*), with the median size and fraction of large colonies remaining stable over 5 hr of flow. The behavior is independent of the total biovolume fraction, showing that aggregation rates are negligible for large colonies. In contrast, the steady median size for small colonies increases with the total biovolume fraction, although still an order of magnitude smaller than division-formed colonies. Significant fragmentation can be seen from a dissipation rate of $\dot{\varepsilon} = 5.8 \, \text{m}^2/\text{s}^3$ (*Appendix 1—figure 1*). For a large value of $\dot{\varepsilon} = 9.6 \, \text{m}^2/\text{s}^3$, the small colonies become the largest fraction in below 5 min (the smallest time resolution in our setup).

## Aggregation measurements with *Microcystis* strain V163

Additional measurements of aggregation of single cells of *Microcystis* strain V163 were performed. The suspension was filtered through a 25 µm filter paper to select mostly single cells. Under a moderate dissipation rate of $\dot{\varepsilon} = 0.019 \, \text{m}^2/\text{s}^3$, the aggregation increased with the total biovolume fraction $\phi$ (*Appendix 1—figure 2*). Under an intense dissipation rate of $\dot{\varepsilon} = 5.8 \, \text{m}^2/\text{s}^3$, no aggregation was observed, and the median size of colonies remained equal to the initial value (near the single-cell size).

## Rate of increase in the average colony size due to pure aggregation

Consider a suspension of colonies subjected only to aggregation, neglecting cell division and fragmentation. In this case, the time derivative of the normalized biovolume distribution $n(l,t)$ is given by *Burd and Jackson, 2009*:

$$\frac{\partial}{\partial t} n(l,t) = \frac{6 \phi \alpha}{\pi} \left[ \frac{1}{2} \int_0^l \beta(l', l_0) \left( \frac{l}{l \, l_0} \right)^{2f-1} n(l',t) \, n(l_0, t) \, dl' - n(l,t) \int_0^\infty \beta(l, l') \frac{1}{l'^f} n(l', t) \, dl' \right],$$

(14)

where $l_o = (l^f - l'^f)^{1/f}$, and the collision frequency kernel is $\beta(l, l')$. The average colony size is defined as $\bar{l} = \int_0^\infty l \, n(l,t) \, dl$. Applying this definition to *Equation 14*, we obtain that the time derivative of $\bar{l}(t)$ follows:

$$\frac{\partial}{\partial t} \bar{l}(t) = \frac{6 \phi \alpha}{\pi} \left[ \frac{1}{2} \int_0^\infty \int_0^l \beta(l', l_o) \, l \left( \frac{l}{l' \, l_o} \right)^{2f-1} n(l', t) \, n(l_o, t) \, dl' \, dl - \int_0^\infty \int_0^\infty n(l,t) \, \beta(l, l') \, l \frac{1}{l'^f} n(l', t) \, dl' \, dl \right].$$

(15)

If we assume that the size distribution remains monodisperse for short times, we can write that $n(l,t) = \delta(l - \bar{l}(t))$. Substituting this hypothesis into *Equation 15* results that:

$$\frac{\partial}{\partial t} \bar{l}(t) = \frac{6 \phi \alpha}{\pi} \left[ \frac{1}{2} \int_0^\infty \int_0^l \beta(l', l_o) \, l \left( \frac{l}{l' l_o} \right)^{2f-1} \delta(l' - \bar{l}) \, \delta(l_o - \bar{l}) \, dl' \, dl - \int_0^\infty \int_0^\infty \delta(l - \bar{l}) \, \beta(l, l') \, l \frac{1}{l'^f} \delta(l' - \bar{l}) \, dl' \, dl \right].$$

(16)

Applying a variable change, the integral can be rewritten as:

$$\frac{\partial}{\partial t}\bar{l}(t) = \frac{6\phi\alpha}{\pi}\left[\frac{1}{2}\int_0^\infty\int_0^\infty \beta(l',l_o)\,l\left(\frac{l}{l'l_o}\right)^{2f-1}\delta(l'-\bar{l})\,\delta(l_o-\bar{l})\,l_o^{f-1}\,(l_o^f+l'^f)^{\frac{1}{f}-1}\,\mathrm{d}l_o\mathrm{d}l' \right.$$
$$\left. -\int_0^\infty\int_0^\infty \delta(l-\bar{l})\,\beta(l,l')\,l\,\frac{1}{l'^f}\,\delta(l'-\bar{l})\,\mathrm{d}l'\mathrm{d}l\right]. \tag{17}$$

Solving the integral and applying the model for the collision kernel in a turbulent shear flow (**Saffman and Turner, 1956**), $\beta(l,l') = 0.163\,(l+l')^3\,\sqrt{\dot{\varepsilon}/\nu}$, results that the rate of increase in the average colony size is:

$$\frac{1}{\bar{l}}\frac{\mathrm{d}\bar{l}}{\mathrm{d}t} = 2.49\,(\sqrt[f]{2}-1)\,\phi\,\alpha\,(\bar{l})^{4-2f}\sqrt{\frac{\dot{\varepsilon}}{\nu}}. \tag{18}$$

In the limit of fractal dimension $f = 2$, the growth rate reduces to:

$$\frac{1}{\bar{l}}\frac{\mathrm{d}\bar{l}}{\mathrm{d}t} = 1.03\,\phi\,\alpha\,\sqrt{\frac{\dot{\varepsilon}}{\nu}}, \tag{19}$$

which is independent of $\bar{l}$.

## Rheological characterization of concentrated colonies of strain V163

A steady-shear (**Appendix 1—figure 9A**) and a large-amplitude-oscillatory-shear (**Appendix 1—figure 9B**) tests were performed in a sample of concentrated colonies of *Microcystis* strain V163. Due to the limited volume of samples available after separation, only one repetition was possible for each test. Before each test, the sample was subject to a pre-shear at $\dot{\gamma} = 10\ \mathrm{s}^{-1}$ for 200 s, followed by 100 s of rest. For the steady-shear test, the minimum transient time was determined in a separate test (40 s for $\dot{\gamma} = 100\ \mathrm{s}^{-1}$) to 200 s for $\dot{\gamma} = 0.001\ \mathrm{s}^{-1}$. For the oscillatory shear, the minimum transient time was determined by the rheometer software. A Herschel-Bulkley fit was applied to the steady-shear data, $\tau = \tau_y + k\,\dot{\gamma}^n$, from which a dynamical yield stress $\tau_y = 4.3 \pm 0.3$ Pa (SD from best fit) was estimated.

## Comparison of hypothesis for the fragment distribution function

Assuming a binary fragmentation (i.e. two fragments per fragmentation event), the fragment size distribution can be written in the form $g_i(l,l') = \delta(l-l_a) + \delta(l-l_b)$, where $l_a$ and $l_b$ are the sizes of the fragments. Two ideal models for binary fragmentation are erosion and equal fragments, described by the following functions:

$$g(l,l') = \begin{cases} \delta(l-1) + \delta(l-\sqrt[f]{l'^f-1}) & \text{(Erosion)}, \\ 2\delta(l-l'/\sqrt[f]{2}) & \text{(Equal fragments)}. \end{cases} \tag{20}$$

Due to a lack of experimental observations of single colony fragmentation events, both models were tested for the colonies of categories $C_1$ and $C_2$. The experimental results for division-formed colonies of *Microcystis* strain V163 were better captured by the erosion hypothesis on category $C_2$ (**Appendix 1—figure 4**). Although the equal-fragment hypothesis recovered the decrease in the median colony diameter, only the erosion hypothesis also recovered the biovolume transfer from large to small colonies. In contrast, the experimental results for aggregated colonies were better captured by the equal-fragment hypothesis on category $C_1$ (**Appendix 1—figure 5**).

### Cone-and-plate shear setup

The turbulent shear was generated by a cone-and-plate shear, with the dimensions depicted in **Appendix 1—figure 7A**. The flow regime can be identified from the average energy dissipation rate $\dot{\varepsilon}$ curve against the angular velocity $\omega$, with a $\sim\omega^2$ dependence in the laminar regime and a $\sim\omega^{5/2}$ dependence in the inertial region (**Appendix 1—figure 7B**; **Sdougos et al., 1984**; **Grad and Einav, 2000**). The Kolmogorov scale $\eta$, which is an estimate of the size of the smallest eddies in a turbulent flow, can be calculated from the relation $\eta = \nu^{3/4}\dot{\varepsilon}^{-1/4}$. The Kolmogorov scale in our results

ranges from $\eta \sim 85$ µm for the lowest dissipation rate ($\dot{\varepsilon} = 0.019\,\mathrm{m}^2/\mathrm{s}^3$) to $\eta \sim 18$ µm for the highest dissipation rate ($\dot{\varepsilon} = 9.6\,\mathrm{m}^2/\mathrm{s}^3$).

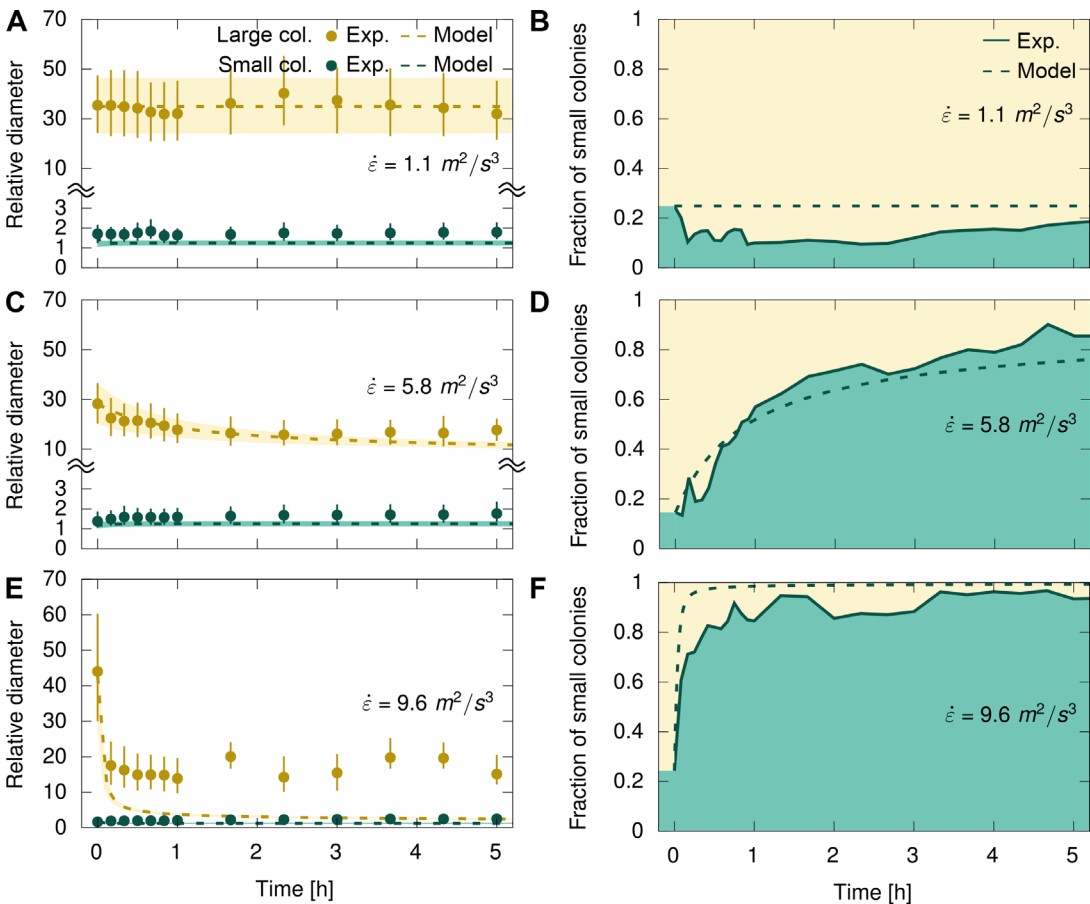

**Appendix 1—figure 1.** Kinetics of the fragmentation of strain V163 colonies under a moderate total biovolume fraction ($\phi = 10^{-4}$) and various values of dissipation rate. The laboratory culture was filtered to select mainly large colonies, and total biovolume fraction was adjusted. Plots in the left column depict the median diameter of each size population as a function of time, where bars and shaded regions indicate limits of 25th and 75th percentiles for the experimental data and model predictions, respectively. Plots in the right column depict the biovolume fraction of small colonies as a function of time. The dissipation rate is (A–B) $\dot{\varepsilon} = 1.1\,\mathrm{m}^2/\mathrm{s}^3$, (C–D) $\dot{\varepsilon} = 5.8\,\mathrm{m}^2/\mathrm{s}^3$, (E–F) $\dot{\varepsilon} = 9.6\,\mathrm{m}^2/\mathrm{s}^3$. Best fit parameters: $\alpha_1 = 0.023$, $S_1 = 0.034$, $q_1 = 4.5$, $S_2 = 31$ $q_2 = 4.1$.

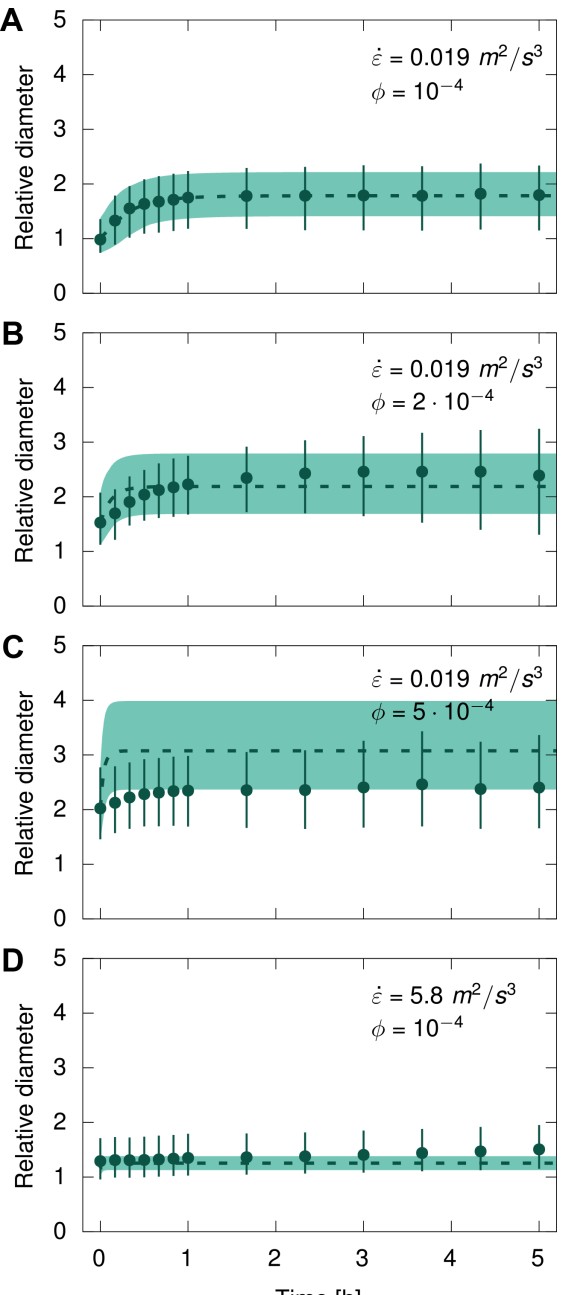

**Appendix 1—figure 2.** Influence of varying dissipation rate and total biovolume fraction on the median diameter of colonies formed by aggregation of single cells of *Microcystis* strain V163. The laboratory culture was filtered to select mostly single cells, and the total biovolume fraction was adjusted before the experiment. (**A**) $\phi = 10^{-4}$ and $\dot{\varepsilon} = 0.019\,\mathrm{m^2/s^3}$, (**B**) $\phi = 2 \cdot 10^{-4}$ and $\dot{\varepsilon} = 0.019\,\mathrm{m^2/s^3}$, (**C**) $\phi = 5 \cdot 10^{-4}$ and $\dot{\varepsilon} = 0.019\,\mathrm{m^2/s^3}$. (**D**) $\phi = 10^{-4}$ and $\dot{\varepsilon} = 5.8\,\mathrm{m^2/s^3}$. Symbols indicate the experimental data, while the lines indicate the predictions from the population model given by *Equation 1*. Bars and shaded regions indicate limits of 25th and 75th percentiles for the experimental data and model predictions, respectively. Best fit parameters: $\alpha_1 = 0.023, S_1 = 0.034$, $q_1 = 4.5$, $S_2 = 31$, $q_2 = 4.1$.

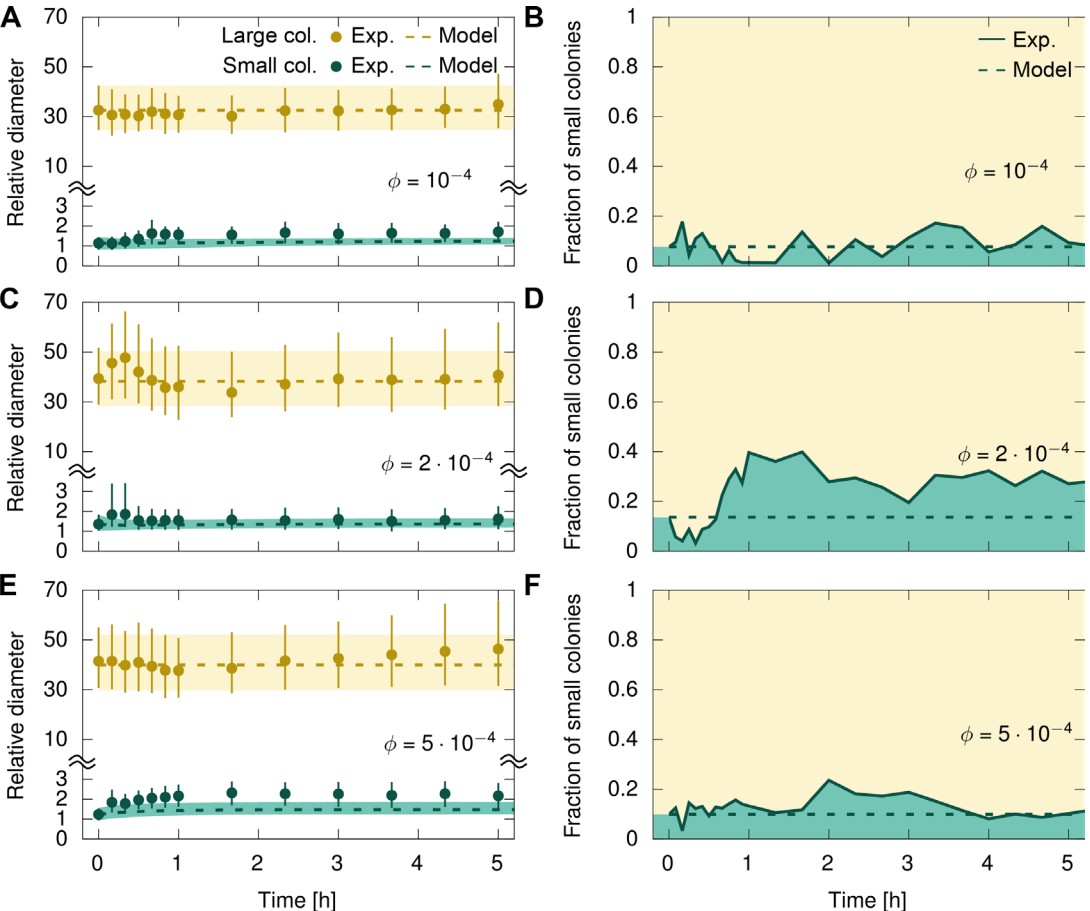

**Appendix 1—figure 3.** Kinetics of the fragmentation of *Microcystis* strain V163 colonies under a moderate dissipation rate ($\dot{\varepsilon} = 0.019\,\mathrm{m^2/s^3}$) and various values of total biovolume fraction. The laboratory culture was filtered to select mainly large colonies, and total biovolume fraction was adjusted. Plots in the left column depict the median diameter of each size population as a function of time, where bars and shaded regions indicate limits of 25th and 75th percentiles. Plots in the right column depict the biovolume fraction of small colonies as a function of time. The total biovolume fraction is (**A–B**) $\phi = 10^{-4}$, (**C–D**)$\phi = 2 \cdot 10^{-4}$, (**E–F**) $\phi = 5 \cdot 10^{-4}$. Best fit parameters: $\alpha_1 = 0.023$, $S_1 = 0.034$, $q_1 = 4.5$, $S_2 = 31$, $q_2 = 4.1$.

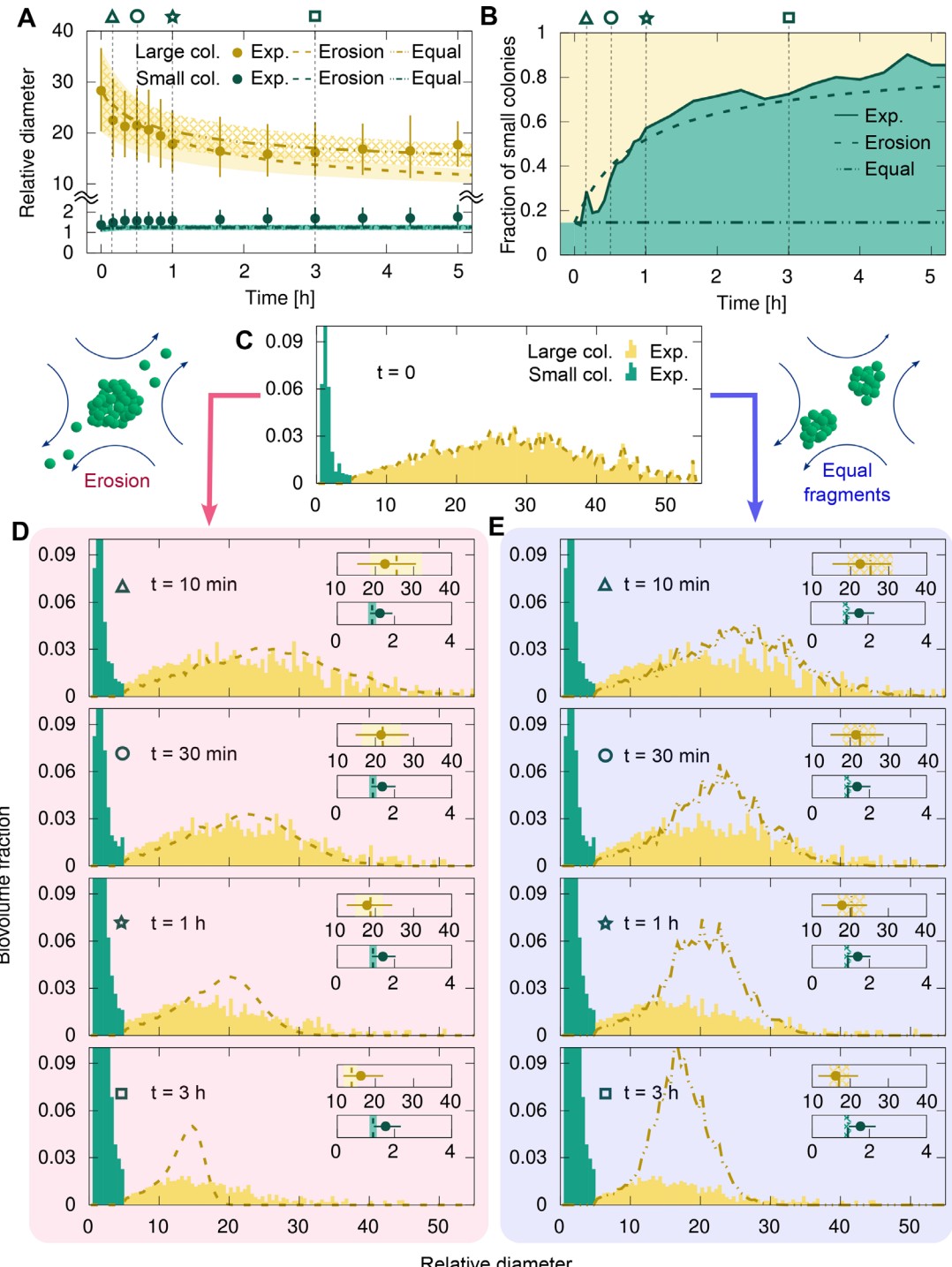

**Appendix 1—figure 4.** Comparison between the erosion and equal-fragment hypothesis for the fragmentation distribution of a suspension of large division-formed colonies. Small aggregates colonies followed the equal-fragment hypothesis. (**A**) Median diameter of small and large colonies as a function of time. Symbols indicate the experimental data, while the lines indicate the predictions from the population model given by **Equation 1**. Both the erosion (dashed) and the equal-fragment (dot-dash) hypothesis recovered well the time behavior of the median diameter. Bars and shaded region indicate limits of 25th and 75th percentiles. (**B**) Biovolume fraction of small colonies (i.e. biovolume of small colonies over the total biovolume) as a function of time. Only the erosion model recovered the transfer from large to small colonies. (**C**) Initial size distribution of colonies expressed as biovolume fraction as a function of the relative colony diameter (normalized by single-cell diameter). Bars indicate the experimental data (large colonies in yellow and small colonies in green), while the lines indicate the model

*Appendix 1—figure 4 continued on next page*

*Appendix 1—figure 4 continued*

predictions for large colonies. Both experiments and models have a bin width of $\Delta l = 0.5$. The model was initialized with the experimental distribution and advanced in time using two fragment distribution functions for the large colonies: (**D**) erosion, in the left column and (**E**) equal fragments, in the right column. Time progresses from top to bottom. Insets in panels D and E display the median diameter of each category, following the symbol notation of panel A. The total biovolume fraction is $\phi = 10^{-4}$ and the energy dissipation rate is $\dot{\varepsilon} = 5.8\,\mathrm{m}^2/\mathrm{s}^3$. Best fit parameters for both models: $\alpha_1 = 0.023$, $S_1 = 0.034$, $q_1 = 4.5$; only the erosion model: $S_2 = 31$, $q_2 = 4.1$; only the equal-fragment model: $S_2 = 33$, $q_2 = 6.3$.

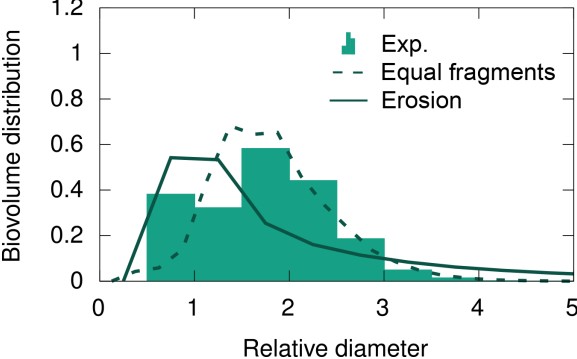

**Appendix 1—figure 5.** Comparison of hypotheses for the fragment distribution of category $C_1$ colonies. Bars show the experimental results for normalized biovolume distribution of filtered single cells of *Microcystis* strain V163 after 1 hr under a moderate dissipation rate ($\dot{\varepsilon} = 0.019\,\mathrm{m}^2/\mathrm{s}^3$) and a total biovolume fraction of $\phi = 10^{-4}$. Lines depict the model predictions using an equal-fragment hypothesis (dashed) and an erosion hypothesis (solid).

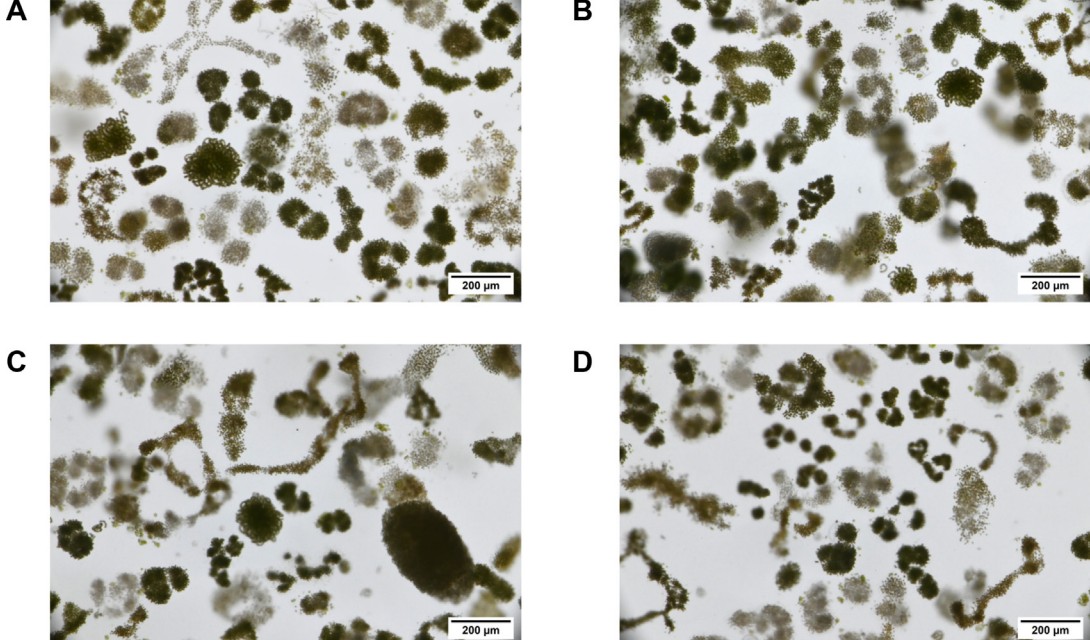

**Appendix 1—figure 6.** Micrographic images of field samples. (**A–D**) Micrographic images of phytoplankton field samples collected from the surface layer of Lake Gaasperplas. *Microcystis* spp. were dominant in the sample.

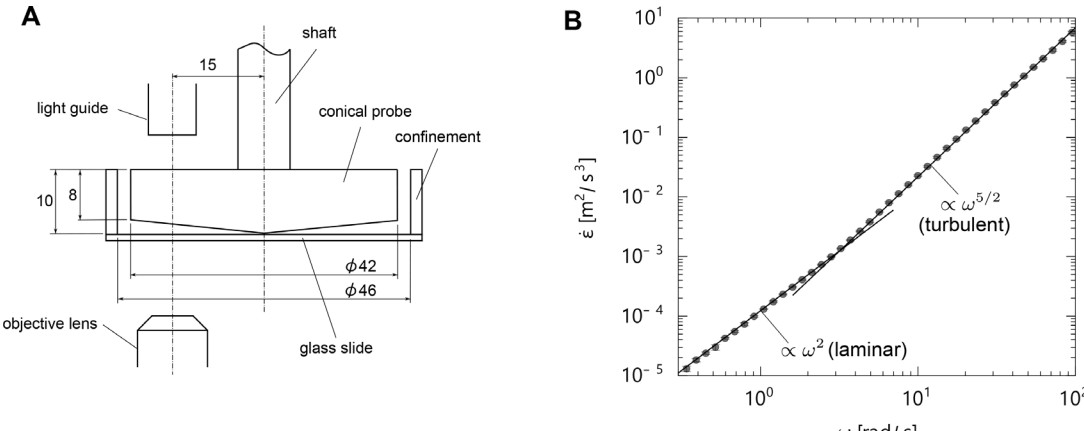

**Appendix 1—figure 7.** Cone-and-plate setup used to generate a turbulent shear in a suspension of *Microcystis* colonies. (**A**) Schematics of the components with the main dimensions indicated in millimeters. (**B**) Average energy dissipation rate as a function of the angular velocity of the conical probe. Bullets indicate the measured data, and the solid lines indicate the best fit of the laminar regime ($\sim \omega^2$) and the inertial regime ($\sim \omega^{5/2}$).

Step 1: The colony size distribution is measured in the cone-and-plate shear setup

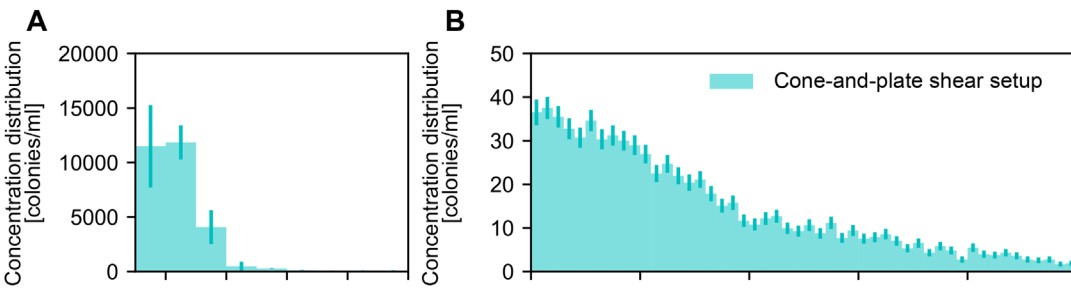

Step 2: The suspension is allowed to sediment inside the cylindrical confinement and the colony size distribution is measured with an inverted microscope

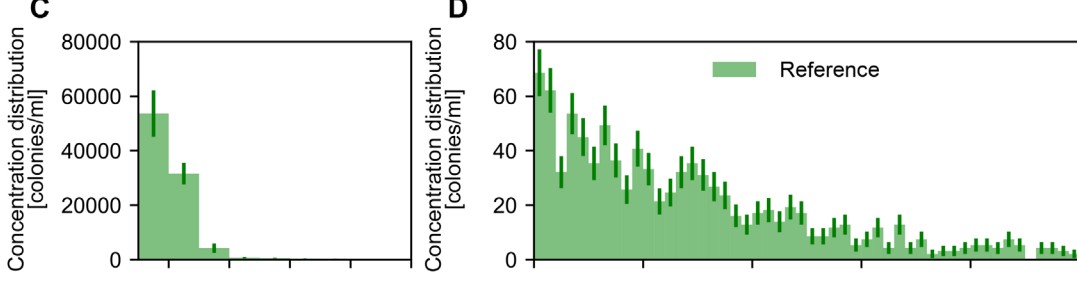

Step 3: The correction function is calculated from the ratio between the reference size distribution and the cone-and-plate size distribution

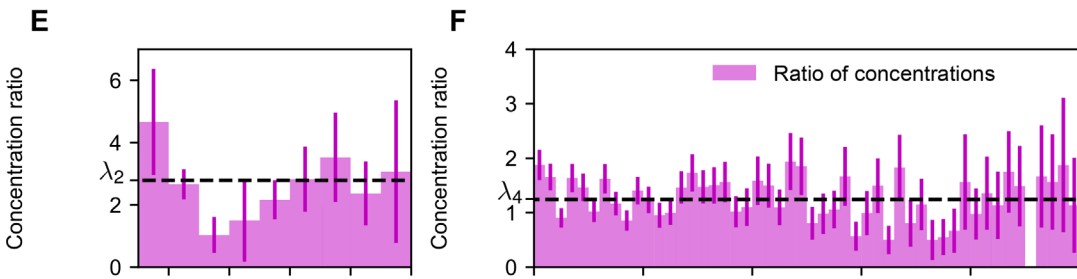

Step 4: The correction function is applied to the colony size distribution measured in the cone-and-plate shear setup, which is then converted to biovolume and normalized

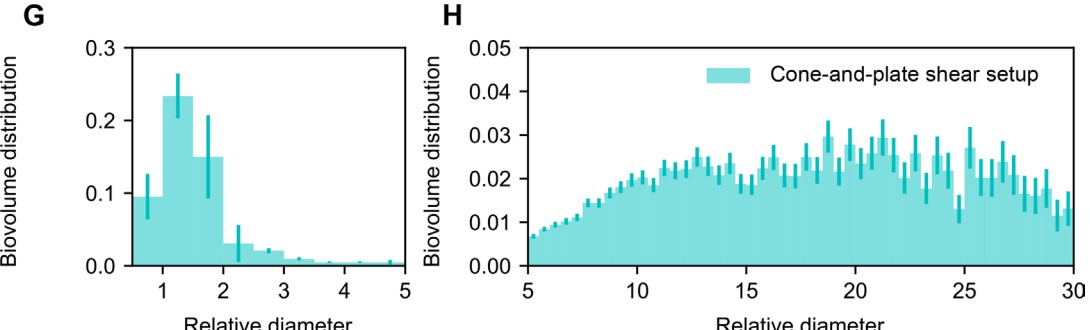

**Appendix 1—figure 8.** Calibration steps for computing the correction function for the biovolume distribution. (**A–B**) Uncorrected concentration distribution of colonies measured in the cone-and-plate shear setup. Number of colonies counted during sampling: $N$=10,776. (**C–D**) Reference concentration distribution measured in the inverted microscope after sedimentation of the colonies. Number of colonies counted during sampling, in panels **C** and **D**: $N = 3066$ and 1455, respectively. (**E–F**) Ratio between the reference and uncorrected concentration distributions. Dashed lines indicate the fitted calibration parameters. (**G–H**) Normalized biovolume distribution measured in the cone-and-plate shear setup after the correction function. Error bars indicate the root mean squared of counting uncertainty and the size measurement uncertainty. Panels **A, C, E, G** display the distributions for small colonies ($l < 5$), while panels **B, D, F, H** display the distributions for large colonies ($l > 5$).

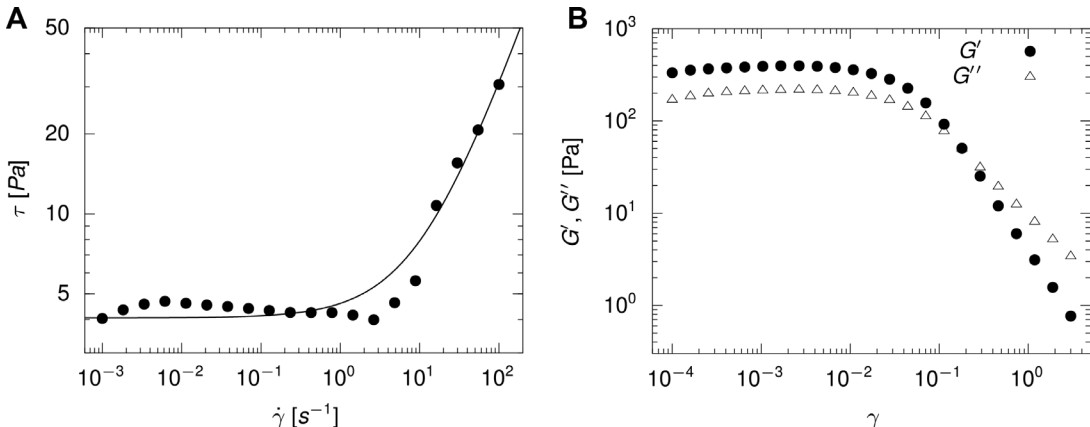

**Appendix 1—figure 9.** Rheology of concentrated colonies of *Microcystis* strain V163. (**A**) Shear stress as a function of the shear rate obtained for a steady-shear test. Solid line indicates a Herschel-Bulkley fit, $\tau = \tau_y + k\dot{\gamma}^n$, where $\tau_y = 4.3 \pm 0.3$ Pa (SD from best fit) is the dynamical yield stress. Other fit parameters are $k = 0.55 \pm 0.13$ Pa and $n = 0.85 \pm 0.05$. (**B**) Storage $G'$ and loss $G''$ moduli as a function of the deformation strain amplitude $\gamma$ for an oscillatory shear with angular frequency of $1$ rad/s.

