## [Editor Report · eLife Assessment]

With the goal of investigating the assembly and fragmentation of cellular aggregates, this manuscript examines cyanobacterial aggregates in a laboratory setting. This quantitative investigation of the conditions and mechanisms behind aggregation is an **important** contribution as it yields a basic understanding of natural processes and offers potential strategies for control. The combination of computational and experimental investigations in this manuscript provides **convincing** support for the role of shear on aggregation and fragmentation.

---

## [Referee Report · Reviewer #2 (Public review)]

Summary:

In this work, the authors investigate the role of fluid flow in shaping the colony size of a freshwater cyanobacterium Microcystis. To do so, they have created a novel assay by combining a rheometer with a bright field microscope. This allows them to exert precise shear forces on cyanobacterial cultures and field samples, and then quantify the effect of these shear forces on the colony size distribution. Shear force can affect the colony size in two ways: reducing size by fragmentation and increasing size by aggregation. They find limited aggregation at low shear rates, but high shear forces can create erosion-type fragmentation: colonies do not break in large pieces, but many small colonies are sheared off the large colonies. Overall, bacterial colonies from field samples seem to be more inert to shear than laboratory cultures, which the authors explain in terms of enhanced intercellular adhesion mediated by secreted polysaccharides.

Strengths:

-This study is timely, as cyanobacterial blooms are an increasing problem in freshwater lakes. They are expected to increase in frequency and severeness because of rising temperatures, and it is worthwhile learning how these blooms are formed. More generally, how physical aspects such as flow and shear influence colony formation is often overlooked, at least in part because of experimental challenges. Therefore, the method developed by the authors is useful and innovative, and I expect applications beyond the presented system here.

-A strong feature of this paper is the highly quantitative approach, combining theory with experiments, and the combination of laboratory experiments and field samples.

Weaknesses:

This study has no major weaknesses. Although the initial part of the introduction seems to imply that fluid flow is the predominant factor in shaping cyanobacterial colony (de)formation, the ensuing discussion is sufficiently nuanced for the reader to understand that the multicellular lifestyle of cyanobacterium Microcystis is shaped by multiple effects, that include bacterial behavior (e.g. which and how much EPS is produced), environmental variables that control cellular aggregation or adhesion and, indeed, fluid flow.

---

## [Author Response]

The following is the authors’ response to the previous reviews

**Public Reviews:**

**Reviewer #1 (Public review):**
Taken altogether, the experimental evidence favors an erosion-dominated process. However, a few minor questions remain regarding the models. Why does the equalfragmentation model predict no biomass transfer between size classes? To what extent, quantitatively, does the erosion model outperform the equal fragments model at capturing the biomass size distributions? Finally, why does the idealized erosion fail to capture the size distribution at late stages in Supplemental Figure S9 - would this discrepancy be resolved if the authors considered individual colony variances in cell adhesion (for instance, as hypothesized by the authors in lines 133-137)? I do not believe these questions curb the other results of the paper.

Our analysis in Figure 2 considers two size classes: small colonies (*l < 5*) and large colonies (*l ≥ 5*). The equal-fragment model predicts that the fracture of a large colony gives rise to two daughter fragments with half the biovolume. For an average colony of *l = 25* in diameter, this corresponds to two daughter fragments with a diameter of *l = 18*, which is still in the large colony class. Sequential fragmentation events would be required to set a biomass transfer to the small size range (*l < 5*). However, the nearly exponential behavior of the fragmentation frequency function (Eq. 5) implies that subsequent fragmentation events are greatly slowed down. Therefore, the equal-fragments model predicts that the biomass transfer from large to small colonies during the first five hours of the experiment is negligible. This is in a sharp contrast with the erosion model, which transfers biomass to the small size class at every fragmentation event. The difference between the two fragmentation models is quantified in Figure 2D, with a negligible change in biomass size distribution for the equal-fragment model (horizontal dash-dotted line) and a strong increase of small colonies for the erosion model (curved dashed line). Hence, it is clear from Figure 2D that the erosion model outperforms the equal-fragment model by capturing the observed shift from large to small colonies. We have now described this more clearly in lines 231-233.

Nevertheless, the performance of the idealized erosion model is limited at late stages (Fig. S9D). We agree with the reviewer that this limitation could potentially be overcome with the introduction of variance in cell adhesion among colonies (as we hypothesized in lines 140142). However, this is not a trivial thing to do, as it would require additional free parameters and reduce the simplicity of the model. Therefore, we chose to restrain our model to the common assumptions of idealized fragmentation models widely used in literature (e.g. references 53-55).

**Reviewer #2 (Public review):**
Especially the introduction seems to imply that shear force is a very important parameter controlling colony formation. However, if one looks at the results this effect is overall rather modest, especially considering the shear forces that these bacterial colonies may experience in lakes. The main conclusion seems that not shear but bacterial adhesion is the most important factor in determining colony size. The writing could have done more justice to the fact that the importance of adhesion had been described elsewhere. This being said, the same method can be used to investigate systems where shear forces are biologically more relevant.

In this work we aimed to investigate the effects of shear forces over a wide range of values, extending beyond the regime of natural lakes into the strong mixing created by technological applications such as the bubble plumes that are applied in several lakes to suppress cyanobacterial blooms. The adhesion force between cells via, e.g., extracellular polysaccharides (EPS) play an essential role by controlling the resistance to shear-driven erosion, which has been quantified in our model by the fitting parameters *Si* and *qi*.

We agree with the reviewer that we have missed some literature on *Microcystis* colony formation via cell aggregation (i.e., cell adhesion), for which we apologize. In our new revision, we have now included several new references [30-34,36] and we now describe the findings of these earlier studies. Specifically, in the Introduction we now pay more attention to the role of cell adhesion by writing (lines 53-60):

“In contrast, cell aggregation (sometimes also called cell adhesion) can promote a rapid increase in colony size beyond the limit set by division rates, and may explain sudden rises in colony size in late bloom periods [26, 30, 31]. Aggregation rates depend on the stickiness of the colonies, which in turn is controlled by the EPS composition, pH, and ionic composition of water [27–29]. In particular, divalent cations such as Ca2+ can bridge negatively charged functional groups in EPS and therefore increase stickiness [32–34]. It has been shown that high levels of Ca2+ enhance cell aggregation in Microcystis cultures [35]. Moreover, cell aggregation can provide a fast defense against grazing [36]. Fluid flow plays an important role in cell aggregation by regulating the collision frequency between cells or colonies [6]. In addition, fluid flow ….”

Furthermore, in the Conclusions we added (lines 374-376):

“A previous study on colony aggregation at high Ca2+ levels observed similar morphological differences in colony formation [35]. There, an initial fast cell aggregation produced a sparse colony structure, followed by a more compact structure of the colonies associated with cell division”

Finally, we would like to clarify a difference in terminology between the reviewer’s comment and our work. The term cell adhesion is commonly used in microbiology to refer to adhesion of cells with a solid substrate. In our work, the adhesion mediated by EPS occurs between free-floating cells and colonies. To avoid any confusion, we chose to refer to this process as cell aggregation, in line with other literature on suspended particles.

**Reviewer #2 (Recommendations for the authors):**
The authors have expanded on the image analysis process but now report substantially different correction factors (λ2 = 2.79 compared to 73.13 in the previous submission; λ3 = 0.52 compared to 13.71 in the previous submission). Could the authors comment on how the analysis changed? These correction factors for N<5 appear particularly relevant for the aggregation experiments presented in Figure 3. For measurements involving only small colonies, as in Figure 3, are these correction factors still valid? In addition, does the timing of image acquisition, i.e. when the colonies are imaged, influence the correction factors applied in this study?

The description of the calibration process was improved in our earlier revision of the manuscript to improve clarity and remove unclear definitions. In the first version, the supplementary equation (S1) for the input variable *Np[i]* was defined as the number of features per frame. This variable is dependent on the frame dimension (2048x2048 px for large colonies, l>5, and 400x400 px for small colonies). We believe that a more suitable input is the concentration distribution, which is normalized by frame area, and therefore invariant to frame dimensions and less prone to misinterpretations. For this reason, we adjusted this definition of *Np[i]* in the revised version of the manuscript, so that it expresses the number of features per frame area (instead of per frame). These changes required the calibration constants, *λ2* and *λ3*, to be updated in the manuscript by a factor of (400 px/2048 px)^2^. This explains why these two calibration constants changed by a factor 0.038. This rescaling of the input variable *Np[i]* and the calibration constants did not affect the final results of our calculations (Figures 2 and 3).

The authors use a moderate dissipation rate to stir the colonies, after which they allow them to sediment. How long were the particles allowed to sediment before measurements were taken? Intuitively, one might expect a greater number of colonies to be detected following sedimentation, yet the authors report only about one third of the colonies in the sedimented state. What accounts for this reduction? Furthermore, if higher shear rates are applied, do the results differ, for instance if particles are lifted further by the shear flow? Some more clarity would help other researchers to perform similar work.

The sedimentation of particles following an initial stir was applied only for creating a reference size distribution, displayed in the supplementary Figures S8-C and D. As one intuitively would expect, a higher concentration of colonies was detected after sedimentation (Fig. S8-C and D) than during the shear flow (Fig. S8-A and B). During all other experiments in our work, the applied dissipation rate was sufficient to ensure a uniform distribution of colonies in suspension throughout the parameter range, as described in lines 461-473.

In the caption of Figure S8 we have reported the number of colonies counted in small subsamples. These numbers are just small subsets of the total number of colonies contained in the entire volume of the cone-and-plate setup. A sub-sample with larger volume was measured during the shear flow in comparison to the sub-sample measured for the sedimented sample, leading to a larger number of counted colonies in panels A and B (N = 10776, combined) compared to panels C and D (N = 3066 and 1455, respectively).

However, when normalized for the volume of the sub-samples, the calculated concentration of colonies is higher for panels C and D (as shown in the graphs). We understand that the earlier caption description of Figure S8 was misleading, for which we apologize. In the revised version, we have adjusted the caption to better describe the quantity:

“Number of colonies counted during sampling …”

Line 797 contains an unfinished edit ("Figure ADD") that should be corrected.

The unfinished edit has been corrected in the newly revised manuscript. Thanks!